# Identifying General Mechanism Shifts in Linear Causal Representations

**Tianyu Chen**[*]   **Kevin Bello**[†‡]   **Francesco Locatello**[◇]   **Bryon Aragam**[†]   **Pradeep Ravikumar**[‡]

[*]Department of Statistics and Data Sciences, University of Texas at Austin
[†]Booth School of Business, University of Chicago
[‡]Machine Learning Department, Carnegie Mellon University
[◇] Institute of Science and Technology Austria

## Abstract

We consider the linear causal representation learning setting where we observe a linear mixing of $d$ unknown latent factors, which follow a linear structural causal model. Recent work has shown that it is possible to recover the latent factors as well as the underlying structural causal model over them, up to permutation and scaling, provided that we have at least $d$ environments, each of which corresponds to perfect interventions on a single latent node (factor). After this powerful result, a key open problem faced by the community has been to relax these conditions: allow for coarser than perfect single-node interventions, and allow for fewer than $d$ of them, since the number of latent factors $d$ could be very large. In this work, we consider precisely such a setting, where we allow a smaller than $d$ number of environments, and also allow for very coarse interventions that can very coarsely *change the entire causal graph over the latent factors*. On the flip side, we relax what we wish to extract to simply the *list of nodes that have shifted between one or more environments*. We provide a surprising identifiability result that it is indeed possible, under some very mild standard assumptions, to identify the set of shifted nodes. Our identifiability proof moreover is a constructive one: we explicitly provide necessary and sufficient conditions for a node to be a shifted node, and show that we can check these conditions given observed data. Our algorithm lends itself very naturally to the sample setting where instead of just interventional distributions, we are provided datasets of samples from each of these distributions. We corroborate our results on both synthetic experiments as well as an interesting psychometric dataset. The code can be found at https://github.com/TianyuCodings/iLCS.

## 1   Introduction

The objective of learning disentangled representations is to separate the different factors that contribute to the variation in the observed data, resulting in a representation that is easier to understand and manipulate [3]. Traditional methods for disentanglement [e.g., 19, 20, 7, 9, 26] aim to make the latent variables independent of each other.

Consider the setting of linear independent component analysis (ICA) [19], that is, the observed variables $X \in \mathbb{R}^p$ are generated through the process $X = GZ$, where $Z \in \mathbb{R}^d$ are *latent* factors, and $G \in \mathbb{R}^{p \times d}$ is an *unknown* "mixing" matrix. Under the key assumption that $Z$ has statistically independent components, and under some additional mild assumptions, landmark results in linear ICA show that it is possible to recover the latent variables $Z$ up to permutation and scaling [13, 19].

---

[1]Emails: tianyuchen@utexas.edu, kbello@cs.cmu.edu

38th Conference on Neural Information Processing Systems (NeurIPS 2024).

However, what if instead of independent sources $Z$ we have a *structural causal model* (SCM, [37, 38]) over them? For instance, if the latent factors correspond to biomarkers in a biology context, or root causes in a root cause analysis context, then we expect there to be rich associations between them. Indeed, this question is central in the burgeoning field of causal representation learning (CRL) [39, 51], where we are interested in extracting the latent factors and causal associations between them given raw data.

Let us look at the simplest CRL setting where the latent variables $Z$ follow a *linear* SCM, that is, $Z = AZ + \Omega^{1/2}\epsilon$, where $A \in \mathbb{R}^{d \times d}$ encodes a directed acyclic graph (DAG), $\Omega$ is a diagonal matrix that controls the scale of noise variances, and $\epsilon$ is some noise vector with zero-mean and unit-variance independent components. In such a case, $Z$ is a linear mixing of independent components $\epsilon$, that is, $Z = B^{-1}\epsilon$, where $B = \Omega^{-1/2}(I_d - A)$ succinctly encodes the SCM and $I_d$ is the identity matrix of dimension $\mathbb{R}^{d \times d}$. We then have $X = GB^{-1}\epsilon$ so that ICA can only recover $BG^{\dagger}$ up to permutation and scaling, which does not suffice to recover the SCM $B$ since the mixing function $G$ is unknown.

Recently, Seigal et al. [40] showed that given the interventional distributions arising from *perfect interventions* on *each* latent variable in $Z$, we can recover the SCM over $Z$ up to permutation. But there are two caveats to this: (a) it is difficult to obtain perfect single-node interventions that only intervene on a single factor in $Z$; and (b) it is difficult to obtain $d$ number of such perfect interventional distributions or environments.

We are interested in the setting where we do not have perfect interventions: we allow for far more general interventions that can quite coarsely change the SCM, namely, *soft* and *hard* interventions, interventions targeting *single* or *multiple* nodes, as well as interventions capable of *adding* or *removing* parent nodes and *reversing* edges. Moreover, we do not need as many as $d$ of these.

Our goal, however, is not to recover the entire SCM over $Z$ but simply to recover those nodes $Z$ that have incurred shifts or changes between the different interventional distributions. This is closely related to root cause analysis [5, 6, 21, 33], which aims to identify the origins of the observed changes in a joint distribution. In addition, understanding the sources of distribution shifts—that is, localizing invariant/shifted conditional distributions—can benefit downstream tasks such as domain adaptation [30], and domain generalization [36, 55].

**Contributions.** Our work sits at the intersection of linear CRL [40, 23] and *direct estimation* of causal mechanism shifts [52, 14]. The key contribution of this work is to show that it is possible to identify the *latent* sources of distribution shifts in multiple datasets while *bypassing* the estimation of the mixing function $G$ and the SCM $B$ over the latent variables, under very general types of interventions. More concretely, we make the following set of contributions:

1. **Identifiability:** We show that we can identify the shifted latent factors even under more general types of interventions. (Section 4.1).

2. **Algorithm:** We also provide an scalable algorithm that implements our identifiability result to infer such shifted latent factors even in the practical scenarios where we are not given the entire coarse interventional distributions but merely finite samples from each (Section 4.2).

3. **Experiments:** We corroborate our results on both synthetic experiments (Section 5.1) as well as an interesting psychometric dataset (Section 5.2).

## 2 Related Work

**Causal representation learning.** In contrast to our setting, which focuses on identifying shifted nodes in the latent representation, existing methods in CRL aim to recover *both* the latent causal graph and the mixing function. Previous works have studied identifiability in various settings, such as latent linear SEMs with linear mixing [40], and with nonlinear mixing [4]; latent nonlinear SEMs with finite degree polynomial mixing [1], and with linear mixing [48]; and nonlinear SEMs with nonlinear mixing [50, 49, 23, 22]. Although these studies ensure the identifiability of causal graphs (up to permutation and scaling ambiguities), they generally rely on the assumption that *each latent variable* is intervened upon in at least one environment, necessitating access to at least $d$ interventional distributions. Moreover, the aforementioned works assume specific types of interventions, such as hard/soft interventions and single-node interventions, and restrict changes in interventional distributions, disallowing edge reversals or the addition of new edges. The most

recent work [23] enables causal representation learning under general interventions in latent linear SEMs with linear mixing. However, this approach still requires the assumption that the number of environments $K$ is at least equal to the number of latent nodes $d$ and that there are at least $\Theta(d^2)$ interventions. If the objective is to detect variables with general mechanism changes across multiple environments—environments that may lack a consistent topological order and sufficient interventions or environments—using existing CRL methods to recover each latent graph becomes overly restrictive or even infeasible. In contrast, we present a more flexible approach, enabling the identification of shifted variables without assuming restrictive interventions per environment or a consistent topological order of the latent graphs.

**Direct estimation of mechanism shifts.** The problem of directly estimating causal mechanism changes *without* estimating the causal graphs has also been explored in various settings in the regime in which the causal variables are observable. Wang et al. [52] and Ghoshal et al. [14] have focused on identifying structural differences, assuming linear SEMs as environments, and proposing methods that take advantage of variations in the precision matrices. More recently, Chen et al. [10] studied this problem for nonlinear additive noise models, assuming that the environments originate from soft/hard interventions and leverage recent work in causal discovery via score matching. Finally, the concept of detecting/localizing feature shifts between two distributions has also been discussed in [27], although from a non-causal perspective. To our knowledge, there is a gap in the literature regarding the study of these objectives when considering latent causal variables. We address this gap by proposing a novel approach for directly detecting mechanism shifts within the latent SCMs.

**Independent component analysis.** The application of independent component analysis (ICA) [12] in the realm of causal discovery has seen significant developments. Linear ICA [19] and its nonlinear counterpart [20] have been instrumental in causal discovery [35, 44, 53] and more recently in causal latent discovery [23]. Beyond these established applications, our work uncovers a novel use of ICA, namely, identifying shifted nodes within the latent linear SCMs.

Given the relevance of ICA for our approach, we briefly recap it next. ICA considers the following setting: $X = W\epsilon$ where $X \in \mathbb{R}^p$, $\epsilon \in \mathbb{R}^d$, $p \geq d$. A key assumption in ICA is that each component of $\epsilon$ is independent. Given only observations of $X$, the goal of ICA is to estimate both $W$ and $\epsilon$. The objective function typically aims to maximize negentropy or non-Gaussianity, with further details given in [19]. The identifiability results of ICA can be summarized as follows.

**Theorem 1** (Theorems 3,4 in [13])**.** *If every component of $\epsilon$ is independent and at most one component is Gaussian distributed, with $W$ being full column rank, then ICA can estimate $W$ up to a permutation and scaling of each column, and $\epsilon$ can be recovered for some permutation up to scaling for each component. Furthermore, as noted in [19], if $\mathbb{E}[\epsilon_i^2] = 1, \forall i \in [d]$, the estimated $W$ and $\epsilon$ will have ambiguities only in permutation and sign. Formally, this means*

$$X = W\epsilon = (WP^TD)(DP\epsilon),$$

*where $P$ is a permutation matrix and $D$ is a diagonal matrix with diagonal entries $\pm 1$. Then, the best estimate given by ICA is $WP^TD$ and $DP\epsilon$.*

## 3  Problem Setting

Consider a random vector $X$ in $\mathbb{R}^p$ that is a linear mixing of $d$ latent variables $Z = (Z_1, \ldots, Z_d)$:

$$X = GZ.$$

Here the latent variables in $Z$ follows a linear SCM [37, 38], that is,

$$Z = AZ + \Omega^{1/2}\epsilon$$

where $A \in \mathbb{R}^{d \times d}$ corresponds to a DAG $\mathcal{G}$ such that $A_{jk} \neq 0$ iff there exists an edge $j \to k$ in the DAG $\mathcal{G}$; $\Omega \in \mathbb{R}^{d \times d}$ is a diagonal matrix with positive entries, and $\epsilon \in \mathbb{R}^d$ is a random vector with independent components with mean zero and variance one, i.e., that $\mathrm{Cov}(\epsilon) = I_d$. Denoting $B = \Omega^{-1/2}(I_d - A)$, we have that:

$$Z = B^{-1}\epsilon.$$

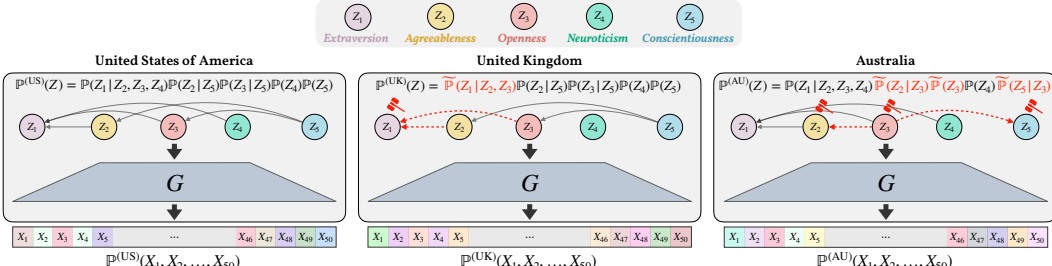

Figure 1: We have 5 *latent* variables $Z$ which in this case relate to personality concepts, and the observations $X$ represent the scores of 50 questions from a psychometric personality test. The latent variables $Z$ follow a linear SCM, while the *unknown* shared linear mixing is a full-rank matrix $G \in \mathbb{R}^{50 \times 5}$. Then, for environment $k = \{\text{US}, \text{UK}, \text{AU}\}$, the observables are generated through $X^{(k)} = GZ^{(k)}$. Here, $\mathbb{P}^{(\text{US})}$ is taken as the "observational" (reference) distribution, and the distribution shifts in $\mathbb{P}^{(\text{UK})}$ and $\mathbb{P}^{(\text{AU})}$ are due to changes in the causal mechanisms of $\{Z_1\}$ and $\{Z_2, Z_3, Z_5\}$, respectively. Finally, the types of interventions are general; for UK, the edge $Z_4 \to Z_1$ is removed and the dashed red lines indicate changes in the edge weights to $Z_1$; for AU, $Z_2$ was intervened by removing $Z_5 \to Z_2$ and *adding* $Z_3 \to Z_2$, while the edge $Z_5 \to Z_3$ was *reversed*, thus changing the mechanisms of $Z_3$ and $Z_5$. Thus, we aim to identify $\{Z_1\}$ and $\{Z_2, Z_3, Z_5\}$.

We assume that we observe $K \geq 2$ generalized interventional distributions that keep the mixing map $G$ fixed but allow for generalized interventions to $Z$. That is, for environment $k \in [K]$ we have,

$$X^{(k)} = GZ^{(k)},$$

where $Z^{(k)} = A^{(k)} Z^{(k)} + (\Omega^{(k)})^{1/2} \epsilon^{(k)}$. Similarly, we have $Z^{(k)} = (B^{(k)})^{-1} \epsilon^{(k)}$, where $B^{(k)} = (\Omega^{(k)})^{-1/2} (I_d - A^{(k)})$.

Notably, we allow generalized interventions that allow for $A^{(k)}$ to be arbitrary, which includes *soft* and *hard* interventions, interventions targeting *single* or *multiple* nodes, as well as interventions capable of *adding* or *removing* parent nodes and *reversing* edges. This contrasts with the existing literature on CRL, where single-node soft/hard interventions are the standard assumption [50, 40, 4, 1]. See Figure 1, for a toy example of what we aim to estimate.

**Remark 1.** *Since we allow for general types of interventions, we can take any of the given environments as the canonical "observational" distribution with respect to which we observe interventions, or simply that we observe $k$ interventions of an unknown observational distribution. This is a clear distinction from the standard setting in CRL [1, 50, 48, 23] which requires to know which environment is a suitable observational distribution.*

To develop our identifiability result and algorithm, we will make additional assumptions on the noise distributions of the linear SEMs.

**Assumption A** (Noise Assumptions). *For any environment $k \in [K]$, let $\epsilon^{(k)} = (\epsilon_1^{(k)}, \dots, \epsilon_d^{(k)})$ be the vector of $d$ independent noises with $\text{Cov}(\epsilon^{(k)}) = I_d$. We have:*

1. *Identically distributed across environments: $\mathbb{P}(\epsilon^{(k)}) = \mathbb{P}(\epsilon^{(k')})$, for all $k' \neq k$.*

2. *Non-Gaussianity: At most one noise component $\epsilon_i^{(k)}$ is Gaussian distributed.*

3. *Pairwise differences: For any $i \neq j$, we have $\mathbb{P}(\epsilon_i^{(k)}) \neq \mathbb{P}(\epsilon_j^{(k)})$ and $\mathbb{P}(\epsilon_i^{(k)}) \neq \mathbb{P}(-\epsilon_j^{(k)})$.*

Assumption A.1 is usually assumed for learning causal models from multiple environments [31, 4]. Assumption A.2 is typically made in causal discovery methods, as detailed in seminal works such as [43, 42, 19, 45] and is considered a more realistic assumption [34]. Assumption A.3 is generally satisfied in a generic sense; that is, when probability distributions on the real line are randomly selected, they are pairwise different with probability one. This assumption is also adopted in [47, 23].

**Assumption B** (Test Function). *We assume access to a test function $\psi$ that maps each noise r.v. to $\mathbb{R}$ s.t. $\psi(\epsilon_i^{(k)}) = \psi(-\epsilon_i^{(k)})$, and $\psi(\epsilon_i^{(k)}) \neq \psi(\epsilon_j^{(k)})$ if $\epsilon_i^{(k)}$ and $\epsilon_j^{(k)}$ are not identically distributed.*

This assumption states that we can access a test function that can help differentiate the noise components. One coarse example is $\psi(y) = \mathbb{P}(|y| \leq 1)$. This assumption is introduced to better understand our method workflow in Section 4, but it is not completely necessary. We discuss how to relax this assumption in Appendix C. Next, we formally define a mechanism shift.

**Definition 1** (Latent Mechanism Shifts)**.** *Let* $\mathrm{PA}(Z_i^{(k)})$ *denote the set of parents of* $Z_i^{(k)}$*. A latent variable* $Z_i$ *is called a latent shifted node within environments* $k$ *and* $k'$*, if and only if:*

$$\mathbb{P}(Z_i^{(k)} \mid \mathrm{PA}(Z_i^{(k)})) \neq \mathbb{P}(Z_i^{(k')} \mid \mathrm{PA}(Z_i^{(k')})).$$

**Remark 2.** *Following Definition 1, $Z_i$ is a latent shifted node between environments $k$ and $k'$ if: (1) The $i$-th rows of $A^{(k)}$ and $A^{(k')}$ are different; (2) $\Omega_{ii}^{(k)} \neq \Omega_{ii}^{(k')}$; or (3) both.*

Definition 1 aligns with those previously discussed in [52, 14, 10], with the key difference that we consider changes in the causal mechanisms of the latent causal variables. However, note that our results also contribute to the setting in which causal variables are observable considering that the mixing function is the identity matrix, that is, $G = I_d$.

# 4 Identifying Shifts in Latent Causal Mechanisms

Following the setup outlined in the previous section, our focus now turns to developing an algorithm to identify latent shifted nodes, given data from multiple environments. First, note that we can write the overall model as a linear ICA problem, where, for any environment $k$, the observation $X^{(k)}$ is a linear combination of independent components $\epsilon^{(k)}$. Specifically, we have

$$X^{(k)} = GZ^{(k)} = G(B^{(k)})^{-1}\epsilon^{(k)}$$

Under the mild conditions given in Assumption A, from classical ICA identifiability results stated in Theorem 1, we can identify $G(B^{(k)})^{-1}$ up to permutation and sign flip. Let $M^{(k)} = B^{(k)}H$ where $H = G^\dagger$. Then, we can only identify $M^{(k)}$ up to permutation and sign flip, which does not suffice to identify the latent SCM encoded in $B^{(k)}$. In sum, what we can only obtain from ICA is

$$\overline{M}^{(k)} = P^{(k)}D^{(k)}B^{(k)}H$$

where $P^{(k)}$ is a permutation matrix, and $D^{(k)}$ is a diagonal matrix with $-1$ or $+1$ on its diagonal. As Seigal et al. [40] points out, it is not possible to identify $B^{(k)}$ further given *generalized interventions*. Our first result is that our present mild assumptions suffice to infer shifted nodes.

**Theorem 2** (Identifiability)**.** *Given access to $K \geq 2$ environments, assume that A and B hold for all environments. Then, all latent shifted nodes are identifiable.*

An interesting facet of our identifiability result is that it is *constructive*. In the next subsection we will provide an explicit algorithm to infer the shifted nodes and prove the main theorem above.

## 4.1 Constructive identifiability

Consider $\epsilon^{(k)} = B^{(k)}HX^{(k)}$ and $\bar{\epsilon}^{(k)} = \overline{M}^{(k)}X^{(k)} = P^{(k)}D^{(k)}B^{(k)}HX^{(k)} = P^{(k)}D^{(k)}\epsilon^{(k)}$, where $\bar{\epsilon}^{(k)}$ and $\overline{M}^{(k)}$ are the output of ICA, which contain the permutation and sign flip ambiguities given by $P^{(k)}D^{(k)}$.

Obtaining a consistent ordering of the noise components across all environments is equivalent to finding $P^{(k)}$. Under Assumption B, and without loss of generality, we consider that $(\epsilon_1^{(k)}, \ldots, \epsilon_d^{(k)})$ are in increasing order with respect to their $\psi$ values. Since $\psi$ is invariant to sign flip, we can calculate $\psi(\bar{\epsilon}_i^{(k)})$ for all $i \in [d]$ and sort the calculated $\psi$ values in increasing order. Let $\overline{P}^{(k)}$ denote the sorting permutation with respect to $\psi$, so that post-sorting, we get $\overline{P}^{(k)}\bar{\epsilon}^{(k)}$.

**Remark 3.** *In Appendix C, we discuss how to relax the assumption on the test function $\psi$.*

**Proposition 1.** $\overline{P}^{(k)} = (P^{(k)})^T$, *i.e.,* $\overline{P}^{(k)}$ *is the inverse permutation of the ICA scrambling.*

From Proposition 1, we thus find that we can unscramble the permutation $P^{(k)}$ by sorting with respect to $\psi$. We get $\overline{P}^{(k)}\bar{\epsilon}^{(k)} = \overline{P}^k P^{(k)}D^{(k)}\epsilon^{(k)} = D^{(k)}\epsilon^{(k)}$ from the above proposition. In other words, we can extract $\widetilde{\epsilon}^{(k)} = D^{(k)}\epsilon^{(k)}$ via $\widetilde{M}^{(k)} = \overline{P}^{(k)}\overline{M}^{(k)} = D^{(k)}B^{(k)}H = D^{(k)}M^{(k)}$ after ICA and sorting by $\psi$.

**Proposition 2.** *Given access to $K \geq 2$ environments, assume that* A *holds. Then, $Z_i$ is identified as a latent non-shifted node between environments $k$ and $k'$ if and only if $M_i^{(k)} = M_i^{(k')}$, where $M_i^{(k)}$ represents the $i$-th row of $M^{(k)}$, and $M^{(k)} = B^{(k)}H$.*

All formal proofs are given in Appendix E. Our next result shows the identifiability of shifted nodes in the unscrambled matrix $\widetilde{M}^{(k)}$.

**Theorem 3.** *$Z_i$ is identified as a non-shifted node if and only if $\widetilde{M}_i^{(k)} = \widetilde{M}_i^{(k')}$ or $\widetilde{M}_i^{(k)} = -\widetilde{M}_i^{(k')}$.*

We can summarize this in the following algorithm, which proves Theorem 2:

- Perform ICA to obtain $\overline{M}^{(k)}$ and $\bar{\epsilon}^{(k)}$ with input $X^{(k)}$.
- Sort by $\psi$ to get the permutation $\overline{P}^{(k)}$ and compute $\widetilde{M}^{(k)} = \overline{P}^{(k)}\overline{M}^{(k)}$ and $\widetilde{\epsilon}^{(k)} = \overline{P}^{(k)}\bar{\epsilon}^{(k)}$.
- Check the condition on $\{\widetilde{M}_i^{(k)} : k \in [K]\}$ to detect if $Z_i$ is a shifted node, as prescribed by Theorem 3.

### 4.2 Finite-sample algorithm

Thus far, we have considered the population setting where we are given the entire interventional distributions. In practice, we are given samples from each of these interventional distributions, so that we have $K$ datasets, one for each of the interventional distributions. The overall algorithm is given next in Alg. 1 (see illustration in Appendix B) with detailed explanations following the algorithm.

---

**Algorithm 1** iLCS: **I**dentifying **L**atent **C**ausal Mechanisms **S**hifts

---

**Require:** Datasets $\{\boldsymbol{X}^{(k)}\}_{k=1}^K$ and threshold $\alpha$ (e.g., 0.5)

  Calculate covariance matrix $\Sigma^{(k)}$ from $\boldsymbol{X}^{(k)}$ for all k

  $d = \max\limits_{k=1,\ldots,K} \mathrm{rank}(\Sigma^{(k)})$

  **for** $k = 1, \ldots, K$ **do**

    //Step 1: $\bar{\epsilon}^{(k)}$ is samples from $\bar{\epsilon}^{(k)}$

    $\bar{\boldsymbol{\epsilon}}^{(k)}, \overline{M}^{(k)} \leftarrow \mathrm{ICA}(\boldsymbol{X}^{(k)}, d)$

    Calculate $\widehat{\psi}(\bar{\boldsymbol{\epsilon}}^{(k)}) = [\widehat{\psi}(\bar{\boldsymbol{\epsilon}}_1^{(k)}), \widehat{\psi}(\bar{\boldsymbol{\epsilon}}_2^{(k)}), \ldots, \widehat{\psi}(\bar{\boldsymbol{\epsilon}}_d^{(k)})]$

    //Step 2

    sorted_idx $\leftarrow$ argsort$(\widehat{\psi}(\bar{\boldsymbol{\epsilon}}^{(k)}))$

    $\widetilde{M}^{(k)} \leftarrow \overline{M}^{(k)}[\text{sorted\_idx}, :]$

  Initialize $S^{(k,k')} = \emptyset$, for all $k \neq k'$

  **for** $i = 1, \ldots, d$ **do**

    **for** $k \neq k'$ **do**

      Calculate $L_i^{k,k'}$

      // Step 3

      **if** $L_i^{k,k'} > \alpha$ **then**

        $S^{(k,k')} \leftarrow S^{(k,k')} \cup \{i\}$

**Ensure:** All latent shifted nodes $S = (S^{(k,k')})_{k,k'}$

---

**Step 1:** We perform ICA with samples from $X^{(k)}$ to extract $\overline{M}^{(k)}$ and samples from $\bar{\epsilon}^{(k)}$.

**Remark 4** (Estimation of $d$.)**.** *One missing component in using ICA in practice is that, along with samples from $X^{(k)}$, we need to input the number of latent nodes $d$, which need to be estimated from samples. Define $\Sigma^{(k)} = \mathbb{E}[X^{(k)}X^{(k)^T}] = G(B^{(k)})^{-1}(B^{(k)})^{-T}G^T$. Since all matrices are full rank, it follows that $d = rank(\Sigma^{(k)})$, where $\Sigma^{(k)}$ can be estimated by the sample covariance matrix. Thus, $d$ can also be estimated by the rank of the sample covariance matrix.*

**Step 2:** We compute the empirical expectation of $\psi$ on samples from $\bar{\epsilon}^{(k)}$, which by law of large number arguments, converges to its population expectation, which is $\psi(\bar{\epsilon}^{(k)})$. We use the sorted order of the empirical expectations to sort the noise components, unscrambling the noise components as earlier, to get $\widetilde{M}^{(k)}$ and samples from $\widetilde{\epsilon}^{(k)}$.

**Step 3:** Here, we explicitly construct a test statistic to check the condition on $\{\widetilde{M}_i^{(k)} : k \in [K]\}$ to detect if $Z_i$ is a shifted node. Note that from our Theorem 3, there is a non-shift node $Z_i$ between environments $k$ and $k'$ if and only if $\widetilde{M}_i^{(k)} = \pm \widetilde{M}_i^{(k')}$. Accordingly, we define a test statistic:

$$L_i^{k,k'} = \frac{\min\{\|\widetilde{M}_i^{(k)} \pm \widetilde{M}_i^{(k')}\|_1\}}{\|\widetilde{M}_i^{(k)}\|_1 + \|\widetilde{M}_i^{(k')}\|_1}$$

It can be seen that $L_i^{k,k'} = 0$ if and only if $\widetilde{M}_i^{(k)} = \pm \widetilde{M}_i^{(k')}$, which implies node $Z_i$ is not shifted between environments $k$ and $k'$. Thus, in step three of the algorithm above, for each coordinate $i \in [d]$, we check if there exists $k \neq k'$ such that $L_i^{k,k'} > \alpha$ for a given threshold $\alpha$. If such a $k \neq k'$ exists, we include $i$ in the list of shifted nodes.

Algorithm 1 is consistent with the ground truth set of shifted nodes as $n$ approaches infinity. Empirical evidence supporting this claim is presented in Figure 2, which shows that with a sufficiently large sample size, all shifted nodes are correctly identified, and the F1 score reaches 1. Further theoretical discussion on the sample complexity of our method can be found in Appendix D.

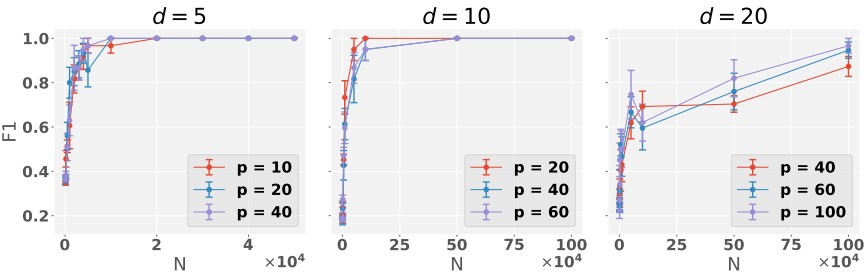

Figure 2: Illustration of the efficacy of our method in accurately identifying latent shifted nodes as the sample size increases, for ER2 graphs. In the first subplot, for a latent graph with $d = 5$ nodes, we examine scenarios with observed dimensions $p = 10, 20, 40$ and plot their corresponding F1 scores against the number of samples $n$. It is observed that the F1 score approaches 1 with a sufficiently large sample size. Detailed experimental procedures and results are discussed in Section 5.

## 5 Experiments

In this section, we investigate the performance of our method in synthetic and real-world data.

### 5.1 Synthetic Data

In our setup, each noise component $\epsilon_i$ is sampled from a generalized normal distribution with the probability density function given by $p(\epsilon_i) \propto \exp\{-|\epsilon_i|^i\}$, where $i = 1, 2, \ldots, d$. In this noise generation process, the noise vector $\epsilon$ adheres to the condition $\psi(\epsilon_i) < \psi(\epsilon_j)$ for all $i < j$ if we choose $\psi(y) = \mathbb{P}(|y| \leq 1)$. Following the methodology similar to that in [40], we start by sampling either an Erdős-Rényi (ER) or Scale-Free (SF) graph with $d$ nodes and an expected edge count of $md$, where $m \in \{2, 4, 6\}$, denoted as $ERm$ or $SFm$. The observed space dimension $p$ is set to $2d$. For each graph, the weights are independently sampled from Unif $\pm [0.25, 1]$ and the diagonal entries of $\Omega$ from Unif$[2, 4]$. In each environment $k$, 15% of the nodes are randomly selected for shifting. The new weights $A_i^{(k)}$ for the shifted node $i$, and the new entries of $\Omega^{(k)}$, specifically $\Omega_{ii}^{(k)}$, are independently sampled from Unif$[6, 8]$. The mixing function $G$ is independently generated from Unif$[-0.25, 0.25]$.

Empirically, we have observed that the following formulation of $L_i^{k,k'}$ leads to improved results:

$$L_i^{k,k'} = \frac{\||\widetilde{M}_i^{(k)}| - |\widetilde{M}_i^{(k')}|\|_1}{\|\widetilde{M}_i^{(k)}\|_1 + \|\widetilde{M}_i^{(k')}\|_1},$$

Table 1: Performance metrics for shifted node detection across various graph configurations, sample sizes $n = 10^6$.

| Graph Type | $p$ | $d$ | Precision | Recall | F1 Score | Time (s) |
|---|---|---|---|---|---|---|
| ER2 | 10 | 5 | 1.000 | 1.000 | 1.000 | 1.23 |
| | 20 | 10 | 1.000 | 1.000 | 1.000 | 3.84 |
| | 40 | 20 | 0.933 | 0.833 | 0.873 | 10.34 |
| | 60 | 30 | 0.680 | 0.700 | 0.689 | 20.06 |
| | 80 | 40 | 0.610 | 0.600 | 0.605 | 30.59 |
| ER4 | 20 | 10 | 1.000 | 1.000 | 1.000 | 3.89 |
| | 40 | 20 | 0.933 | 0.933 | 0.933 | 9.39 |
| | 60 | 30 | 0.617 | 0.600 | 0.607 | 30.83 |
| | 80 | 40 | 0.610 | 0.617 | 0.613 | 32.08 |
| SF2 | 10 | 5 | 0.900 | 0.900 | 0.900 | 1.64 |
| | 20 | 10 | 1.000 | 1.000 | 1.000 | 3.84 |
| | 40 | 20 | 0.807 | 0.833 | 0.817 | 15.85 |
| | 60 | 30 | 0.730 | 0.750 | 0.739 | 22.12 |
| | 80 | 40 | 0.667 | 0.667 | 0.667 | 30.29 |
| SF4 | 20 | 10 | 1.000 | 1.000 | 1.000 | 3.13 |
| | 40 | 20 | 0.967 | 0.900 | 0.927 | 15.12 |
| | 60 | 30 | 0.725 | 0.700 | 0.711 | 29.79 |
| | 80 | 40 | 0.539 | 0.533 | 0.535 | 30.84 |

where $|\widetilde{M}_i^{(k)}|$ denotes the element-wise absolute value of the vector $\widetilde{M}_i^{(k)}$. We will utilized the new formula of $L_i^{k,k'}$ to detect shifts in the following experiment. Then we explore sample sizes $n$ from $500$ to $10^6$, using the observed samples $X^{(k)}$ as input. The parameter $\alpha$ is set to $0.2$ for $d \leq 10$ and $0.5$ for higher dimensions, reflecting the increased complexity in estimating larger dimensional latent graphs and thus necessitating a higher tolerance for $L_1$ norm differences in detecting shifted nodes. For each setting, we independently generate 10 datasets and take the average of the metrics. The results for $n = 10^6$ are shown in Table 1, and the asymptotic consistency results for specific $p$ values are illustrated in Figure 2. In addition to the causal representative setting, our method can also directly identify mechanism shifts in a fully observed setting, where $G = I$. We further compare our method's results in this fully observed setting against the baseline DCI [52], which addresses direct mechanism shifts in linear settings. The results of this comparison are provided in Appendix F, demonstrating that our method outperforms DCI in most settings.

## 5.2 Psychometrics Data

We evaluate our method using a dataset related to the Five Factor Model, also known as the Big Five personality traits [16, 15, 32]. This model is a widely accepted framework, comprising five broad dimensions that encapsulate the diversity of human personality traits. These dimensions are *Openness to Experience, Conscientiousness, Extraversion, Agreeableness*, and *Neuroticism*.

The dataset utilized in our study was gathered through an interactive online personality test available on OpenPsychometrics.org, a nonprofit endeavor aimed at educating the public about psychology while collecting data for psychological research[1]. This dataset encompasses responses to 50 questions, with 10 questions dedicated to each of the five personality dimensions. Participants responded to each question on a scale from 1 to 5. Additionally, the dataset includes demographic information, such as race, age, gender, and country, comprising a total of 19,719 observations.

**Question formalization and data processing.** In this study, we hypothesize the existence of 5 latent nodes, each representing one of the five personality dimensions, believed to be causally related. The score responses to the 50 questions form our observed space. Our main goal is to determine whether variations in personality dimensions can be observed across genders, thus treating gender as one environment ($K = 2$). Additionally, we investigate potential personality shifts across countries, selecting the US and UK for analysis due to they have the most observations in our dataset. The only preprocessing step undertaken involves the removal of observations with missing values and the

---

[1]The data can be downloaded via the link: https://www.kaggle.com/datasets/lucasgreenwell/ocean-five-factor-personality-test-responses/data

normalization of data to fit within the $[0, 1]$ range, achieved by adjusting according to the maximum and minimum values observed. The research question we have formalized in this study is not derived from any data competition. It aligns with interests explored in existing psychological literature [25, 8, 46, 29], yet our investigation is distinguished by a unique analytical framework.

**Labeling latent nodes.** Prior to detecting shifted nodes, it is essential to assign semantics to each node. This process involves conducting interventions on each component of the noise vector to aid in labeling the latent nodes. Given that the noise components are distinct for each latent node, labeling the noise effectively equates to labeling the latent nodes.

Initially, we apply ICA to the data for males, followed by getting post-sorting $\widetilde{M}^{male}$ and $\widetilde{\epsilon}^{male}$ as outlined in our methodology. Subsequently, we perform interventions on each noise component, setting each to 0 sequentially, and then re-mixing the intervened noise vector using $(\widetilde{M}^{male})^\dagger$. By examining the impact of these interventions on the observation space — specifically, identifying which question scores undergo significant changes — we can assign appropriate semantic labels to each latent node index. For instance, nullifying the first column of $\widetilde{\epsilon}$ and remixing the intervened noise with $(\widetilde{M}^{male})^\dagger$ alters the score distribution in a manner that reveals the semantic domain affected by the first noise component. An example of assigning the label *Agreeableness* to a latent node is depicted in Figure 3. By applying the same process to all noise components, we are able to assign semantic labels *Openness, Conscientiousness, Extraversion*, and *Neuroticism* to the remaining latent nodes. More detailed experiment results are shown in Section G.

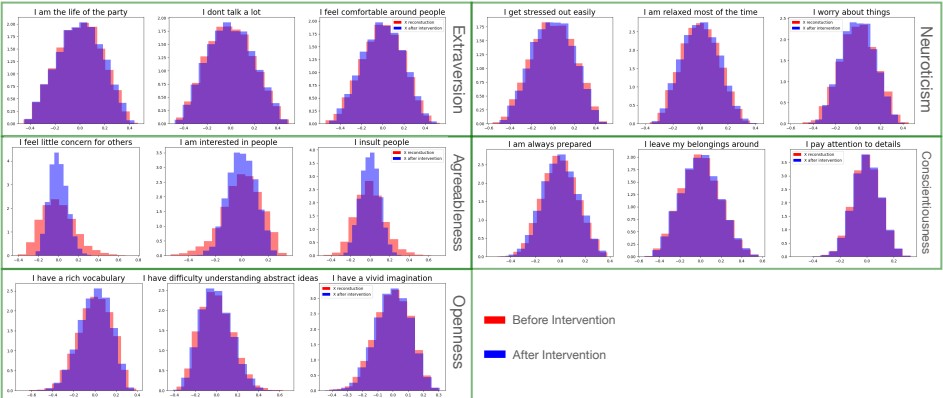

Figure 3: We apply an intervention to the first column of $\epsilon$ and then use $(\widehat{M}^{male})^\dagger$ for remixing. The first row of the resulting histograms represents scores for 5 out of the 10 questions related to the Extraversion personality dimension. Subsequent rows display histograms for 5 questions from each of the other four personality dimensions, as indicated at the right end of each row. The red distribution represents the scores before the intervention on the noise, while the blue distribution corresponds to scores after the intervention. Overlapping areas are shown in purple. Notably, the intervention on the first column of $\epsilon$ alters the distribution in the observed space, specifically affecting the scores for questions related to the *Agreeableness* personality dimension, whereas distributions for other dimensions remain unchanged. Consequently, we can label the first noise component as corresponding to *Agreeableness*.

**Shifted nodes detection.** To identify shifted personality dimensions across gender, we computed $L_i^{male,female}$ for each latent node, obtaining values of $\{0.074, 0.0497, 0.078, 0.638, 0.633\}$. Setting a tolerance threshold $\alpha = 0.5$ to accommodate real data estimation variances, we observed that the last two nodes exhibit significantly higher $L_i^{male,female}$ scores, surpassing $\alpha$, and thus are considered shifted. These nodes correspond to the labels *Neuroticism* and *Extraversion*. Consistent with existing psychological literature, women have been found to score higher in *Neuroticism* than men [25, 8, 46, 29], while men scored higher in the Activity subcomponent of *Extraversion* [8]. This discovery aligns with the findings in psychology literature. To further validate our method's effectiveness, a similar analysis was conducted across countries, comparing the UK and the US, which have the most observations in our dataset. The computed $L_i^{US,UK}$ for each latent node was $\{0.302, 0.258, 0.109, 0.189, 0.088\}$. All values fell below $\alpha$, indicating no latent node shifts between

these two countries. This finding is also in agreement with existing studies that personality exhibits stability across countries and cultures [25, 24, 11].

## 6   Concluding Remarks

In this study, we demonstrated that latent mechanism shifts are identifiable, up to a permutation, within the framework of linear latent causal structures and linear mixing functions. Furthermore, we introduced an algorithm, grounded in ICA, designed to detect these shifts. Our method offers a broader applicability to various types of interventions compared to CRL framework. Unlike shift detection methods where node variables are directly observable, our approach extends to scenarios where latent variables remain unobserved. A promising future direction consists of adapting our methodology to nonlinear transformations, which could address more complex, practical challenges, such as identifying latent mechanism shifts in real-world image data.

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

# SUPPLEMENTARY MATERIAL
# Identifying General Mechanism Shifts in Linear Causal Representations

## A    Limitations and Broader Impacts

Limitations of this work include the need to relax the noise assumption and to consider similar settings under nonlinear mixing functions. These are promising directions to explore in the CRL field. The broader impact of this work is that CRL methods can be used to identify mechanism shifts and determine root causes, which can be utilized in the biological field to find disease genes or biomarkers. Currently, the negative impacts of this method are not clear.

## B    Illustration of our algorithm

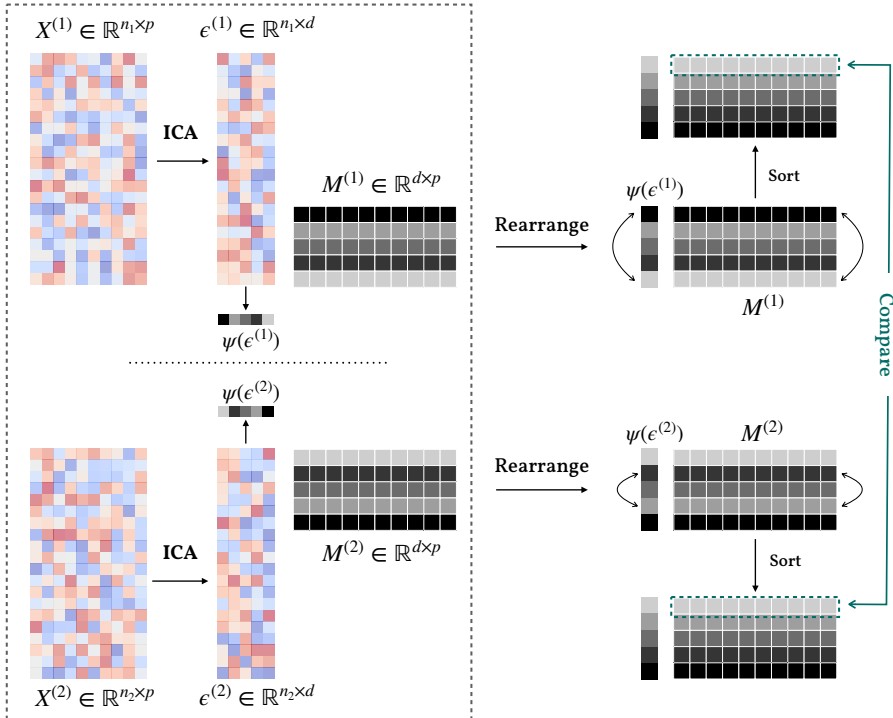

Figure 4: Overview of our method: For each context $k$, given the data $\boldsymbol{X}^{(k)}$, our method involves three main steps. First, we apply ICA to each dataset to estimate $\boldsymbol{\epsilon}^{(k)}$ and $M^{(k)}$. Second, we calculate $\psi(\epsilon^{(k)}) = \{\psi(\epsilon_1^{(k)}), \psi(\epsilon_2^{(k)}), \ldots, \psi(\epsilon_d^{(k)})\}$ for each noise component, sort these components in increasing order, and correspondingly arrange the rows of $M^{(k)}$. Third, we compare the sorted rows of $M^{(k)}$ to identify the shifted nodes.

## C    Discussion on Test Function

In Assumption B, we assume that there exists a test function $\psi$ and that we can access it. Here we discuss ways to relax it. Recall that in Section 4.1, $\psi$ is utilized to sort the noise component $\bar{\epsilon}^{(k)}$ to ensure that the post-sorting noise vector $\widetilde{\epsilon}^{(k)}$ has a consistent order across all environments.

An alternative approach to achieve this is to use distribution matching. We take the noise vector in the first environment as a reference and align all other noise vectors post-sorting with the reference vector. To do this, we can use a distribution distance metric $D$. First, define a signed permutation space as

$$S_d = \{S = PD \mid P \text{ is a permutation matrix, } D \text{ is a diagonal matrix with } D_{ii} \in \{-1, 1\}\}$$

Then, solve the optimization problem:

$$\min_{S \in S_d} D(\bar{\epsilon}^{(1)}, S\bar{\epsilon}^{(k)})$$

where $D$ can be any distribution distance, such as Kullback-Leibler divergence. In Assumption A, we assume pair-wise different noise component, thus the optimization questions have minimums value 0 if and only if each noise component of $\bar{\epsilon}^{(1)}$ and $S\bar{\epsilon}^{(k)}$ have the same distribution, thus help us align the noise component order. We solve this optimization problem for each environment $k \geq 2$, thus obtaining $\overline{P}^{(k)}$. All following steps in our algorithm remain the same when using this alternative approach.

One small gap remains: even though all post-sorting noise vectors have a consistent order with $\bar{\epsilon}^{(1)}$, $\bar{\epsilon}^{(1)}$ is not the ground truth order of $\epsilon^{(1)}$. This ambiguity cannot be eliminated, consistent with the nature of the CRL method, and is the same with other CRL methods, such as [40, 28]. Fortunately, the ground truth order is not so important in practice. What people mainly care about is the semantic label for each latent node. Some CRL generative models, such as [54], may be helpful for performing fake interventions and manually assigning semantic labels. However, this is beyond the scope of this paper, and we will not discuss it further.

Even though the distribution matching optimization method offers greater flexibility, it is computationally expensive. First, note that the cardinality $|S_d| = d! \times 2^d$, which represents a vast search space when $d$ is large. Furthermore, calculating $D(\cdot, \cdot)$ is generally computationally intensive. For example, the KL method requires density estimation, and the Maximum Mean Discrepancy (MMD) method necessitates the computation of pairwise distances among samples. These challenges render this alternative difficult to implement. Consequently, we opt to use the $\psi$ function to facilitate efficient sorting, but it may need carefully design.

## D  Discussion on Sample Complexity

The sample complexity of our method must be considered from two perspectives: one involves using ICA to estimate $\bar{\epsilon}^{(k)}$ and $\overline{M}^{(k)}$, and the other pertains to utilizing $\bar{\epsilon}^{(k)}$ and a test function to sort the rows of $\overline{M}^{(k)}$. Since the sorting step depends on the choice of test function, we assume for simplicity that $\overline{M}^{(k)}$ is already sorted by the ground truth order. Thus, we only focus on the asymptotic behavior of $\widetilde{M}^{(k)}$, which closely relates to the properties of the ICA estimator.

There are various algorithms for solving ICA [18, 17, 41]; each algorithm exhibits different asymptotic statistical properties. If we apply the findings in Auddy and Yuan [2], we assume that the estimated ICA unmixing function has the following statistical accuracy:

**Theorem 4.** *If the sample size $n \geq g(d, \delta)$, then with probability at least $1 - h(n, d, \delta, \epsilon)$, we have:*

$$l(\widetilde{M}_i^{(k)} - M_i^{(k)}) \leq C \cdot p(d, n)f(\delta),$$

*where $\widetilde{M}_i^{(k)}$ represents the $i$-th row of the estimated unmixing function $M^{(k)}$, $C$ is a constant, and $p$, $f$, $g$, and $h$ are known functions. For instance, in Auddy and Yuan [2], $p(d, n) = \sqrt{d/n}$ and $f(\delta) = \sqrt{\log(1/\delta)}$. Here, $l$ denotes the loss function, and the $L_2$ norm can serve as an option.*

Under this theorem, for two environments $k$ and $k'$, if node $i$ does not shift, we have:

$$||\widetilde{M}_i^k - \widetilde{M}_i^{k'}||_2 \leq ||\widetilde{M}_i^k - M_i||_2 + ||\widetilde{M}_i^{k'} - M_i||_2 \leq 2 \cdot C \cdot p(d, n)f(\delta)$$

with a probability of at least $1 - 2h(n, d, \delta, \epsilon)$. Thus, by setting the threshold $\alpha$ as $2 \cdot C \cdot p(d, n)f(\delta)$, we can control the false discovery rate to be at most $2h(n, d, \delta, \epsilon)$. A similar sample complexity theorem can be extended to cases involving more than two environments, as long as the statistical properties of the ICA solution are known.

# E  Detailed Proofs

## E.1  Proof of Proposition 2

**Lemma 1.** *Under problem setting, for any $x, y \in \mathbb{R}^{d \times 1}$, the equation $x^T H = y^T H$ holds if and only if $x = y$.*

*Proof.* Given that $G$ possesses full column rank, it follows that $H = G^\dagger$ has full row rank. Consequently, the null space of $H^T$ is $\{0\}$. Therefore, if $x^T H = y^T H$, it implies $H^T(x - y) = 0$. This leads to the conclusion that $x - y = 0$, which in turn implies $x = y$. $\qquad\square$

*Proof of Proposition 2.* Recall that $B^{(k)} = (\Omega^k)^{-\frac{1}{2}}(I_d - A^{(k)})$. Since $A^{(k)}$ is a weighted adjacency matrix, its diagonal entries are zero. Thus,

$$B_{ij}^{(k)} = -\left(\Omega_{ii}^{(k)}\right)^{-\frac{1}{2}} A_{ij}^{(k)} \quad \text{if} \quad i \neq j,$$

$$B_{ii}^{(k)} = \left(\Omega_{ii}^{(k)}\right)^{-\frac{1}{2}} \quad \text{if} \quad i = j.$$

Under Definition 1, if node $Z_i$ is shifted, it implies either 1) $\Omega_{ii}^{(k)} \neq \Omega_{ii}^{(k')}$, 2) $A_i^{(k)} \neq A_i^{(k')}$, or 3) both conditions hold. In scenarios 1) and 3), $B_{ii}^{(k)} \neq B_{ii}^{(k')}$, resulting in $B_i^{(k)} \neq B_i^{(k')}$. In scenario 2), while $\Omega_{ii}^{(k)} = \Omega_{ii}^{(k')}$, there exists a $j \in [d]$ such that $A_{ij}^{(k)} \neq A_{ij}^{(k')}$, leading to $B_i^{(k)} \neq B_i^{(k')}$. If node $Z_i$ is not shifted, then $A_i^{(k)} = A_i^{(k')}$ and $\Omega_{ii}^{(k)} = \Omega_{ii}^{(k')}$, implying $B_i^{(k)} = B_i^{(k')}$. Therefore, $Z_i$ is shifted if and only if $B_i^{(k)} \neq B_i^{(k')}$. According to Lemma 1, $B_i^{(k)} \neq B_i^{(k')}$ if and only if $B_i^{(k)} H \neq B_i^{(k')} H$, which is equivalent to $M_i^{(k)} \neq M_i^{(k')}$.

In conclusion, $Z_i$ is shifted if and only if $M_i^{(k)} \neq M_i^{(k')}$. $\qquad\square$

## E.2  Proof of Theorem 3

**Lemma 2.** *Under problem setting, it is not possible for an intervention on the latent node $Z_i$ to result in $M_i^{(k)} = -M_i^{(k')}$.*

*Proof.* We prove this by contradiction. Suppose that $M_i^{(k)} = -M_i^{(k')}$. According to Lemma 1, this would imply $B_i^{(k)} = -B_i^{(k')}$. However, we know $B^{(k)} = (\Omega^{(k)})^{-1}(I_d - A^{(k)})$ where $A^{(k)}$ is the weight matrix for a DAG. Since $A_{ii}^{(k)} = 0$, it follows that $B_{ii}^{(k)} = (\Omega^{(k)})_{ii}^{-1}$. Therefore, both $B_{ii}^{(k)}$ and $B_{ii}^{(k')}$ are positive. It is impossible for $B_i^{(k)}$ to be equal to $-B_i^{(k')}$. Consequently, the scenario where $M_i^{(k)} = -M_i^{(k')}$ cannot occur. $\qquad\square$

*Proof of Theorem 3.* Recall from the data generation process that

$$M^{(k)} X^{(k)} = \epsilon^{(k)}.$$

When input $X^{(k)}$ to ICA, we have $\overline{M}^{(k)} = P^{(k)} D^{(k)} M^{(k)}$ and $\bar{\epsilon}^{(k)} = P^{(k)} D^{(k)} \epsilon^{(k)}$. Without loss of generality, we assume that $\epsilon^{(k)}$ is ordered increasingly with respect to $\psi$. Thus, post sorting with respect to $\psi$, we eliminate the ambiguity of $P^{(k)}$, and we get $\widetilde{M}^{(k)} = D^{(k)} M^{(k)}$ and $\widetilde{\epsilon}^{(k)} = D^{(k)} \epsilon^{(k)}$.

We are now ready to prove that $Z_i$ is not shifted if and only if $\widetilde{M}^{(k)} = \pm \widetilde{M}^{(k')}$. This immediately implies that if $Z_i$ is not shifted, then $M_i^{(k)} = M_i^{(k')}$, thus satisfying $\widetilde{M}^{(k)} = \pm \widetilde{M}^{(k')}$.

If $\widetilde{M}^{(k)} = \pm \widetilde{M}^{(k')}$, there are two cases: $M_i^{(k)} = M_i^{(k')}$ or $M_i^{(k)} = -M_i^{(k')}$. We prove in Lemma 2 that the scenario $M_i^{(k)} = -M_i^{(k')}$ cannot exist. The only surviving situation is $M_i^{(k)} = M_i^{(k')}$, which indicates that $Z_i$ is not shifted.

$\qquad\square$

# F   Experiments on Synthetic Data Compared with DCI

As described in Section 5.1, instead of generating the mixing function $G$ from $\text{Unif}[-0.25, 0.25]$, we set $G = I$, such that $X = Z$ and $Z$ can be directly observed. In this setup, finding general interventions in linear causal representations reduces to identifying general interventions in linear SEM, a setting for which the existing method DCI [52] is designed. Table 2 presents the performance comparison between our method and DCI under these conditions, demonstrating that our method outperforms DCI in most cases.

| Graph Type | $d$ | Method | Precision | Recall | F1 |
|------------|-----|--------|-----------|--------|-----|
| ER 2 | 5 | DCI | 0.60 | 0.60 | 0.60 |
| | | Ours | 0.80 | 0.80 | 0.80 |
| | 10 | DCI | 0.87 | 1.00 | 0.92 |
| | | Ours | 1.00 | 1.00 | 1.00 |
| | 15 | DCI | 0.74 | 1.00 | 0.84 |
| | | Ours | 0.66 | 1.00 | 0.78 |
| ER 4 | 10 | DCI | 0.83 | 1.00 | 0.89 |
| | | Ours | 1.00 | 1.00 | 1.00 |
| | 15 | DCI | 0.71 | 1.00 | 0.81 |
| | | Ours | 0.62 | 0.93 | 0.73 |
| SF 2 | 5 | DCI | 0.70 | 0.80 | 0.73 |
| | | Ours | 1.00 | 1.00 | 1.00 |
| | 10 | DCI | 0.67 | 1.00 | 0.79 |
| | | Ours | 1.00 | 1.00 | 1.00 |
| | 15 | DCI | 0.65 | 1.00 | 0.78 |
| | | Ours | 0.70 | 0.93 | 0.78 |
| SF 4 | 5 | DCI | 0.60 | 0.60 | 0.60 |
| | | Ours | 0.80 | 0.80 | 0.80 |
| | 10 | DCI | 0.77 | 1.00 | 0.85 |
| | | Ours | 1.00 | 1.00 | 1.00 |
| | 15 | DCI | 0.56 | 0.93 | 0.68 |
| | | Ours | 0.67 | 1.00 | 0.79 |

Table 2: Comparison of Precision, Recall, and F1 scores for different graph types, $d$ values, and methods between our method and DCI.

# G   Additional Information on Real Data

This section provides detailed information on the procedures employed in analyzing the real dataset.

**Preprocessing**   The initial dataset comprised 19,719 observations, which can be downloaded from https://www.kaggle.com/datasets/lucasgreenwell/ocean-five-factor-personality-test-responses/data. In the preprocessing phase, any observation with a missing value in any column was excluded, leaving a total of 19,710 observations for further analysis. Subsequently, we applied max-min value normalization to the scores of each question, ensuring that all scores were normalized to fall within the range $[0, 1]$. This normalization step is crucial for achieving uniformity in the data scale, thereby facilitating accurate analysis and comparison across the dataset.

**Labeling the Noise**   To derive meaningful psychological insights, it is crucial to assign semantic labels to all latent nodes. Given that the noise components are pairwise distinct and unique to the latent node $Z_i$, we consider intervening on each noise component, then remixing and observing the changes in the observational space. This approach enables us to assign semantic labels to both the noise components and their corresponding latent nodes. We utilize observations from the male dataset as the reference context, which comprises 7,603 observations. Following the initial step of our method, we obtain the sorted $\widetilde{M}^{male}$ and $\widetilde{\epsilon}^{male}$. The mixing function $G$ is derived from $(\widetilde{M}^{male})^\dagger$.

To identify the semantic label for the first component of $\widetilde{\epsilon}$, we set its corresponding noise vector component to 0, effectively nullifying the first component of $\widetilde{\epsilon}^{male}$. This intervention yields an estimated noise matrix samples from $\widetilde{\epsilon}_{inv}^{male}$, denoted as $\widehat{\epsilon}_{inv}^{male}$. The intervened reconstruction, $\boldsymbol{X}_{inv}^{male} = G(\widehat{\epsilon}_{intv}^{male})^T$, and the original score distribution, $\boldsymbol{X}^{male} = G(\widehat{\epsilon}^{male})^T$, allow us to compare question scores pre- and post-intervention. Figure 7 plots these distributions, revealing significant shifts for questions pertaining to the *Agreeableness* dimension, with minimal impact on other scores, thereby identifying the first noise component as *Agreeableness*. This process is replicated for the second through fifth columns of $\epsilon^{male}$, with results illustrated in Figures 9, 8, 5, and 6. Each plot demonstrates that interventions result in significant distribution changes for questions related to a single personality dimension, with negligible effects on others. Consequently, we label these noise components as *Openness*, *Conscientiousness*, *Extraversion*, and *Neuroticism*, respectively. These labels will be used for all the following analysis.

**Shifted Nodes Detection**   We then applied our method to data from the male and female contexts. The calculated $L_i^{male,female}$ values are $\{0.074, 0.0497, 0.078, 0.638, 0.633\}$. Based on these results, we identify shifts in the last two personality dimensions, specifically labeled as *Extraversion* and *Neuroticism*. Additionally, we conducted a comparative analysis of personality dimensions between the US and UK, which have 8,753 and 1,531 observations, respectively. The computed $L_i^{US,UK}$ values are $\{0.302, 0.258, 0.109, 0.189, 0.088\}$, indicating that no latent node is considered as having undergone shifts between these two countries.

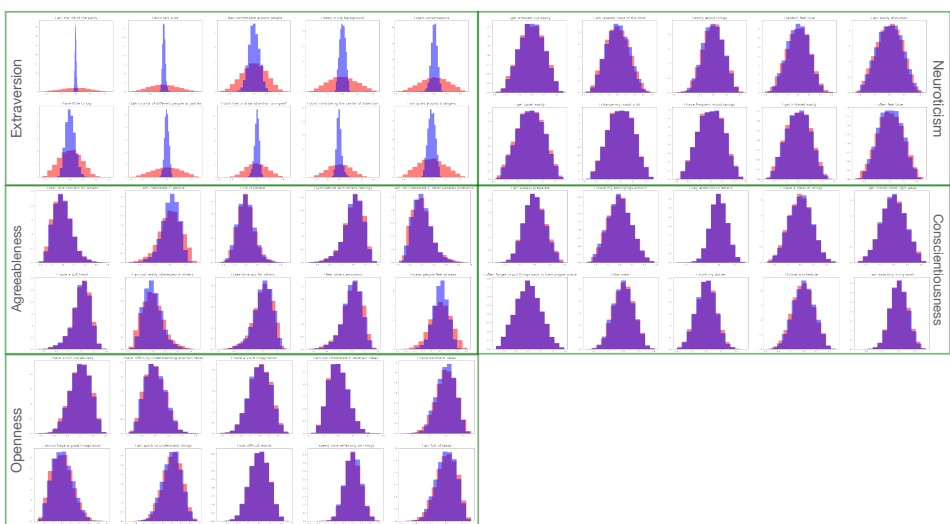

Figure 5: Intervention on the fourth component of the noise vector and subsequent re-mixing generate a new observed space — a new score distribution. Notably, only *Extraversion* exhibits significant changes after intervention, leading us to label the fourth component of the noise vector (after sorting) as *Extraversion*.

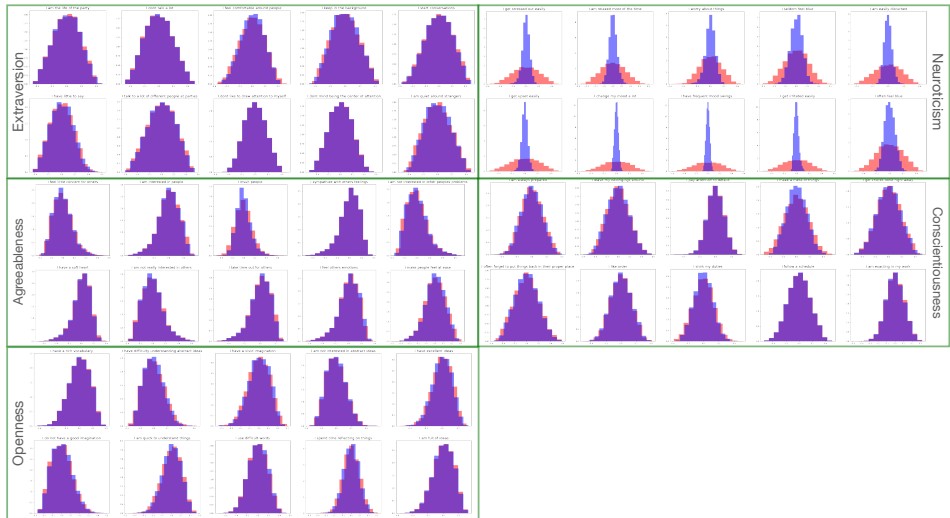

Figure 6: Intervention on the fifth component of the noise vector and subsequent re-mixing generate a new observed space — a new score distribution. Notably, only *Neuroticism* exhibits significant changes after intervention, leading us to label the fifth component of the noise vector (after sorting) as *Neuroticism*.

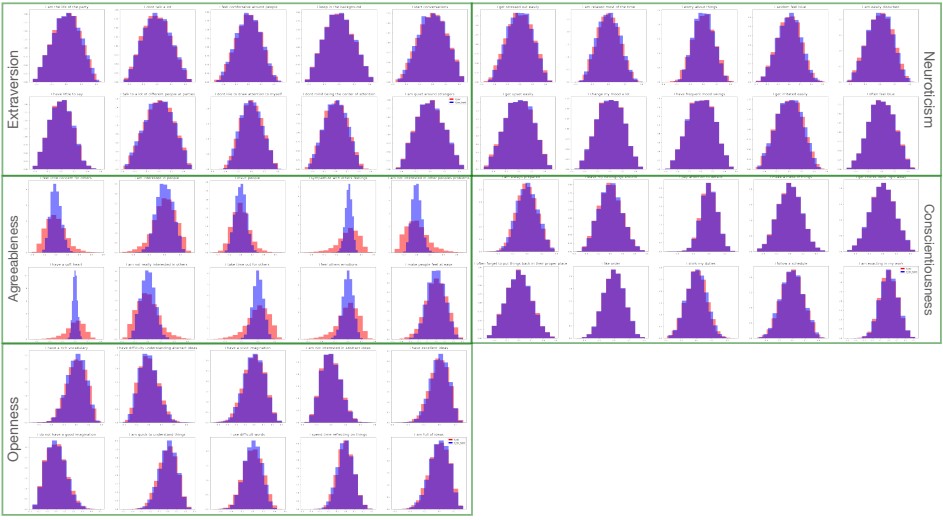

Figure 7: Intervention on the first component of the noise vector and subsequent re-mixing generate a new observed space — a new score distribution. Notably, only *Agreeableness* exhibits significant changes after intervention, leading us to label the first component of the noise vector (after sorting) as *Agreeableness*.

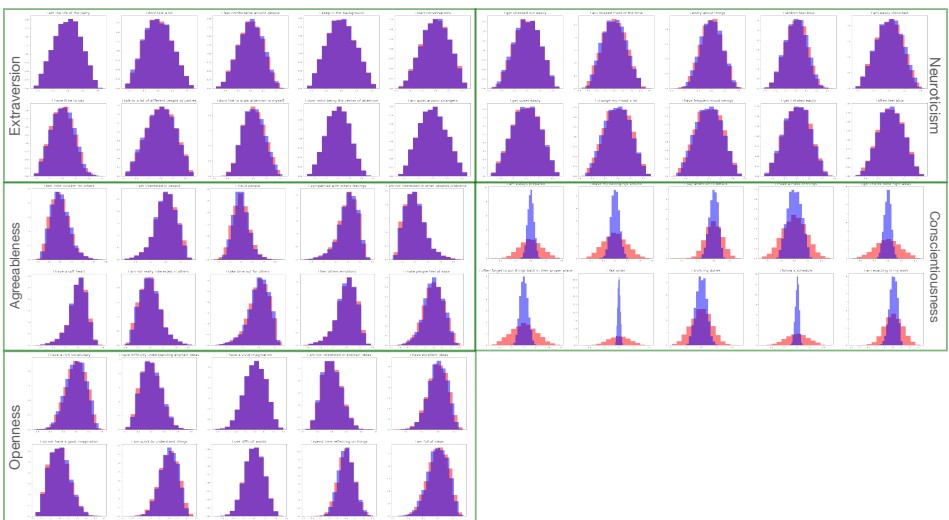

Figure 8: Intervention on the third component of the noise vector and subsequent re-mixing generate a new observed space — a new score distribution. Notably, only *Conscientiousness* exhibits significant changes after intervention, leading us to label the third component of the noise vector (after sorting) as *Conscientiousness*.

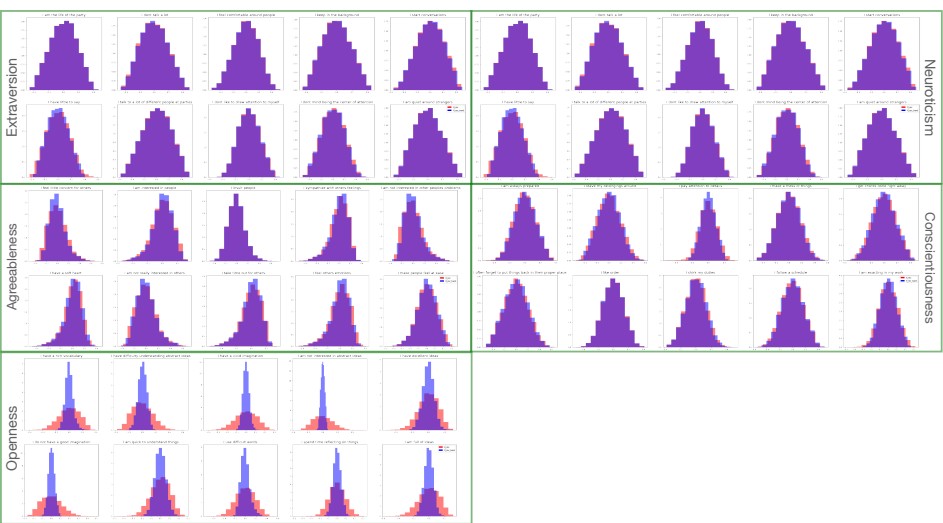

Figure 9: Intervention on the second component of the noise vector and subsequent re-mixing generate a new observed space — a new score distribution. Notably, only *Openness* exhibits significant changes after intervention, leading us to label the second component of the noise vector (after sorting) as *Openness*.

