# OpenReview forum: "Identifying General Mechanism Shifts in Linear Causal Representations"
_NeurIPS.cc/2024/Conference — NeurIPS 2024 poster_

### Official Review · Reviewer_JGz2 · 2024-07-09

**Soundness:** 2
**Presentation:** 4
**Contribution:** 2
**Rating:** 5
**Confidence:** 5

**Summary:**

Recently, causal representation learning has drawn a lot of attention in the representation learning area, where it considers a causal relation among the latent generative factors. This work is under the setting of linear causal representation learning. Data $X$ comes from a linear mixing $X=GZ$ of the unknown latent factors $Z$, and latent factors $Z$ follow a linear SCM. They are trying to estimate the causal mechanism's shifting (causally changed) node, under the more general/relax intervention. They claim that ‘it is possible to identify the latent sources of distribution shifts while bypassing the estimation of the mixing function and the SCM (causal relation matrix). The experiments were provided to justify their setting.

**Strengths:**

Pros:
1. This paper is very well-written. It's pretty clear and easy to understand. The reviewer enjoys reading it.
2. They are considered a more relaxed setting of intervention. It's an interesting idea where you directly find the shifted node.

**Weaknesses:**

Cons:
1. The first issue is the main motivation of this work. Their idea is to bypass the estimation of the mixing function $G$ and the SCM $B$ (causal relation matrix). However, the reviewer believes this is the main motivation for causal representation learning. Why would we want a shift node, if we can't even identify the causal relationship in CRL? Could the author please clarify the point here?

2. The reviewer believes the strong assumption limits the significance of this work. The `access to a test function (Assumption B)' is a really strong assumption that seems not realistic. Since you got the permutation ambiguity fixed. And I believe you also fixed the sign ambiguity in your assumption. Combined with the bounded variance you basically got an ICA with only scaling ambiguity. What is the difference between this work and the classical linear ICA result?

3. The reviewer would like to see the clarification of the motivation and connection between linear ICA and this work. For linear ICA, we can get mixing function $M$ and the independent sources $\epsilon$ in an unsupervised way, with an identifiability guarantee. But for causal representation cases, the $M$ decomposes into $G$ and $B$. The natural/intuitive idea is to identify those factors with certain assumptions. Could the author provide any intuition for this?

The reviewer would love to raise the score if these questions were resolved in the discussion session.

**Questions:**

Besides the three questions I mentioned in the cons section, I do have another question here, but it does not lower the marks of this work, just discussion.
1. This question is not an issue for this work since it seems many reference works use interventional settings. But it would be interesting if the author could discuss this. Could the author clarify why we must use interventional data? Is there any alternative way since the interventional data is kind of a semi-supervised strong assumption? I know there is a paper about the 'Challenging Common Assumptions in unsupervised CRL', but is there any other way we can avoid using interventional data?

---

> ### Author Rebuttal · Authors · 2024-08-06
>
> Thank you for your efforts on reviewing our paper! Below, we address your concerns.
>
> ### From weakness
>
> 1. We provided some motivation from Line 34 to Line 38 in the paper, and also included a toy example in Figure 1. Additionally, we provided a Psychometrics data application (Section 5.2) to reflect the motivation and practical significance of our method. In addition, we would like to provide more potential applications here to illustrate in which scenarios one might be interested in localizing shifted nodes.
>
>     For example, examining the mechanism changes in the gene regulatory network structure between healthy individuals and those with cancer may provide insights into the genetic factors contributing to specific cancers. Within biological pathways, genes could regulate various target gene groups depending on the cellular environment or the presence of particular disease conditions [1,2]. In the analysis of EEG signals [3], it is of interest to detect neurons or different brain regions that interact differently when the subject is performing different activities. All these questions can be formulated as lozalizing shifted nodes.
>
>     The complementary question of finding shifted nodes is to find the shared non-shifted nodes, which can also be used for social science analysis. For instance, we may have data from different countries and different ethnicities but aim to identify the potential causes that influence the years of education pupils receive [4]. Based on the applications in these fields, I believe that estimating shifted nodes in CRL is an interesting and valuable topic.
>
>     [1] Hudson, N. J., Reverter, A., \& Dalrymple, B. P. (2009). A differential wiring analysis of expression data correctly identifies the gene containing the causal mutation. PLoS computational biology.
>
>     [2] Pimanda, J. E., Ottersbach, K., Knezevic, K., Kinston, S., Chan, W. Y., Wilson, N. K., ... \& Göttgens, B. (2007). Gata2, Fli1, and Scl form a recursively wired gene-regulatory circuit during early hematopoietic development. Proceedings of the National Academy of Sciences.
>
>     [3] Sanei, S., \& Chambers, J. A. (2013). EEG signal processing. John Wiley \& Sons.
>
>     [4] Ghassami, A., Salehkaleybar, S., Kiyavash, N., \& Zhang, K. (2017). Learning causal structures using regression invariance. Advances in Neural Information Processing Systems, 30.
>
>
> 2. Assumption B is included for better presentation and understanding of our method's workflow, but it is not necessary for our  identifiability results. As mentioned in line 148 and Appendix C, there are alternatives to this assumption. With estimated noise samples, we can perform distribution matching to achieve a consistent order of noise components. Therefore, this assumption is not necessary.
>
>     Regarding sign and scaling ambiguity, no, we do not fix sign ambiguity in our assumptions, as ICA inherently has sign ambiguity. In Theorem 3, we prove that the unmixing matrix estimated from the ICA method, even with sign ambiguity, will not affect our estimation of shifted nodes. Additionally, in Theorem 1, it is proven that under certain assumptions, the ICA solution only has permutation and sign ambiguity, so there should be no scaling ambiguity.
>
>
>     ICA is an important component of our method. The main contribution of our paper is to prove that the ICA solution can help us detect the shifted latent nodes. We also provide the identifiability result for the algorithm designed based on ICA. We believe this is not trivial since we are solving a new problem with theoretically backed methods.
>
> 3. Thank you for the question. We refer you to the global response for a detailed answer to this question. In short, in the "inadequate" intervention data setting (small number of intervention environments, inadequate number of interventions, and general intervention setting), estimating $G$ and $B$ from $M$ is impossible. Our method bypasses the step of estimating $B$ but can still estimate shifted nodes with fewer assumption constraints.
>
>
> ### From questions
>
> Thank you for the question. Assuming interventional distributions is a modeling choice, which may or may not hold in practice. Our setting is concerned with identifying the sources of distribution changes (shifted nodes) across two or more populations. In this case, one formal way to characterize distribution changes is through modeling interventions in a causal graph. We do not think one *must* assume this but we believe it can be a reasonable assumption in multiple scenarios.
>
>
> ---
> We hope our answers will make you feel more positive about our work. Please let us know any follow-up questions!

---

> > ### Comment · Reviewer_JGz2 · 2024-08-11
> > **Response to the author**
> >
> > W1 The reviewer thanks the author for the clarification and agrees that it is of certain value for finding the shifted nodes. however, it does not fully resolve my question. The reviewer respectfully argues that the shift node setting is not aligned with the fundamental motivation of causal representation learning, which is finding the causal graph/mechanism.
> >
> >
> > W2 The author answered my question about the strong assumption of the test function. The reviewer was satisfied with their explanation about relaxation.
> >
> > W3 For the motivation, the author referenced paper [1] is probably the closest linear model setting. It showed an interventional-based linear model. They showed the number of interventions needed for their method but that seems not a strict lower bound. The reviewer respectfully disagrees with the author's claim 'in the "inadequate" intervention data setting, estimating $G$ and $B$ from $M$ is impossible'. Please correct me if I was wrong, and I would also like to know if there is any reference proving the strict lower bound of intervention needed for identifiability.

---

> > > ### Author Response · Authors · 2024-08-13
> > >
> > > Dear reviewer, thanks for your response. We are glad we were able to solve your concern about the assumption on the test function. Next we address your remaining questions.
> > >
> > > **W1**
> > >
> > > We are not sure why the reviewer is concerned about our setting not being aligned with CRL? Nowhere in our paper we claim that our goal is to solve the CRL problem. Our paper considers the CRL setting in the sense that our model of the data generating process follows that of the linear CRL setting, that is, the latent variables follow an structural causal model, but we explicitly mention the goal of our work in our contributions and the paragraph before (Lines 55-72).
> > >
> > > **W3**
> > >
> > > Thank you for your question. Indeed, several papers discuss the strict lower bound on the number of interventions needed for causal structure recovery. For example, in [1], Theorem 2 demonstrates that with perfect interventions on each single node across different environments, the causal structure can be estimated up to a permutation. Moreover, Proposition 5 indicates that if the number of interventions is fewer than $d$ (where $d$ is the number of latent nodes), the causal structure becomes non-identifiable. Additionally, Appendix B of the same paper shows that if the interventions are soft rather than perfect, Theorem 2 may no longer hold. The most recent advancement in this area is found in [2], which relaxes the hard intervention assumption but still requires at least $d$ environments and $\Theta(d)$ soft interventions.\
> > > These identifiability limitations in CRL w.r.t. the number of environments and the type of interventions are precisely what motivate the search for alternative goals that are still useful in practice, such as identifying mechanism shifts, all while bypassing full identification of the causal structures.
> > >
> > >
> > > [1] Squires, Chandler, et al. "Linear causal disentanglement via interventions." International Conference on Machine Learning. PMLR, 2023.
> > >
> > > [2] Jin, Jikai, and Vasilis Syrgkanis. "Learning causal representations from general environments: Identifiability and intrinsic ambiguity." arXiv preprint arXiv:2311.12267 (2023).

---

> > > > ### Comment · Reviewer_JGz2 · 2024-08-13
> > > > **Response to the author**
> > > >
> > > > The reviewer thanks the author for the clarification. While I'm not a big fan of the scope of identifying mechanism shifts compared to the classical task of identifying causal relation graphs in CRL, I'm raising my score to 5 considering the potential value in corresponding downstream applications in their setting.

---

### Official Review · Reviewer_33zq · 2024-07-11

**Soundness:** 3
**Presentation:** 3
**Contribution:** 3
**Rating:** 6
**Confidence:** 4

**Summary:**

This work studies the nontrivial problem of causal representation learning from the perspective of mechanism shifts within the latent SCM. Specifically, the authors relax existing restrictive assumptions in interventional causal representation learning, such as data generated from single-node perfect interventions and the number of environments necessary to facilitate identifiability and show that it is possible to identify the latent nodes that shift between environments/distributions from more general soft/hard and add/reverse interventions given access to fewer environments than the number of causal variables. Furthermore, the authors develop a practical algorithm to recover the latent sources attributable to the distribution shift and evaluate their method on synthetic data and a psychometric dataset. The main contribution in this work seems to be the use of a test function to score the noise factors learned from ICA to construct a sorted permutation matrix, which is used to construct the new unmixing matrix for scrambling the independent SCM noise variables.

**Strengths:**

- The theoretical result and intuition of identifying latent sources of distribution shift is interesting and is a step toward more feasible CRL for real-world application. Most work focuses on a supervised discriminative setting for studying distribution shifts. This work stands out in being one of the first to identify latent sources of distribution shifts.
- The empirical evaluation is extensive and considers real-world datasets to evaluate the proposed CRL algorithm. The results from the Psychometrics dataset suggest that the algorithm proposed is capable of identifying the latent shifts from the data distribution to a great degree and in line with human interpretation. The test statistic proposed to quantify the degree of distribution shift between nodes w.r.t unmixing matrix is practical.
- Generalizing the class of interventions for interventional CRL to identify only shifted nodes is a useful result for real-world CRL, where interventions may be multi-node and more complex.

**Weaknesses:**

- This work considers linearity in both the mixing and the SCM, which can be somewhat of a restricting assumption in practice.
- Proposition 2 and Theorem 3 seem to contradict each other. By the current logic, the two statements imply that both shifted and non-shifted nodes require the same condition of the row corresponding to the variable index in the unmixing matrix to be invariant across environments (w/ sign flip). This should also be made clear in Section 4.2 in Step 3.
- Minor points
    - In the caption of Figure 1, for the UK environment, it should be the edge Z_4 → Z_1 is removed instead of Z_5 → Z_1 removed.

**Questions:**

- Proposition 2 and Theorem 3 are contradicting. What are the criteria for a node to be identified as a shifted node? Do you mean that if the ith row of the unmixing matrix is **different between environments**, then node i is a shifted node? This seems to be the case when looking at the proof of Proposition 2 in the appendix. I would appreciate it if the authors could provide some clarification on this.
- Do the authors have any intuition about the setting where noise factors are correlated and the iid assumption of noise across environments is violated? In this scenario, we would no longer have the standard linear ICA result to build off of.

**Limitations:**

Limitations are discussed in the appendix.

---

> ### Author Rebuttal · Authors · 2024-08-06
>
> We thank the reviewer's valuable suggestions and for recognizing the novelty of our work. We next address the reviewer's concerns.
>
> ### From weaknesses
>
> * Please refer to our global response regarding linear models.
>
> * Thanks for pointing this out. We apologize for the typo in Proposition 2, line 187. It should state that "$Z_i$ is identified as a **not** shifted latent node between $k$ and $k'$ if and only if $M_i^{(k)} = M_i^{(k')}$", and Theorem 3 is correct. We now believe Proposition 2 and Theorem 3 are consistent after correction.
>
>     Additionally, in Section 4.3, Step 3, line 213, it should read "$Z_i$ is a **non-shift** node between $k$ and $k'$ if and only if $\widetilde{M_i}^{(k)} = \pm \widetilde{M_i}^{(k')}$", and in line 215, it should state "there is **no shift** in node $Z_i$ if and only if $\widetilde{M_i}^{(k)}=\pm \widetilde{M_i}^{(k')}$". We will correct these in the revision.
> Thank you again for your careful review.
>
> * Correct. Thanks for pointing this out. We will correct it in the revision.
>
> ### From questions
>
> * As we pointed above, there are some a couple of typos in Proposition 2 and Section 4.2, Step 3. After correction, as you suggested, the illustration of our method should be consistent. A node $Z_i$ is a shifted node if and only if $M_i^{(k)} \neq M_i^{(k')}$, which means that the $i$-th row of $M^{(k)}$ and $M^{(k')}$ are different, or $\widetilde{M_i}^{(k)} \neq \pm \widetilde{M_i}^{(k')}$.
>
> * Great question! This is a non-trivial question that makes for an exciting future direction. It is important to note that the correlated noise assumption impacts the identifiability of $B^{(k)}$. Its effect on the ICA solution would come secondary. For our setting and for CRL in general, it is crucial that $B^{(k)}$ is unique to avoid identifiability issues. If we assume the noise component is correlated, it implies that $\Omega$ is not a diagonal matrix, and it is also possible that $\Omega$ is not full rank. Given that $\Omega^{1/2}B = I - A$, if $\Omega$ is not full rank, then $B$ is not unique.
>
> ---
> We hope our answers will make you feel more positive about our work. Please let us know any follow-up questions!

---

> > ### Comment · Reviewer_33zq · 2024-08-10
> >
> > I thank the authors for the clarifying response. The authors have done a good job of answering my questions. The problem addressed is quite interesting with significant implications in downstream distribution shift generalization. Furthermore, I believe the theoretical identifiability results and practical algorithm are of interest to the CRL community. Since this work helps to bridge the gap between the theory and practice of CRL, I am increasing my score.

---

> > > ### Author Response · Authors · 2024-08-11
> > >
> > > Dear reviewer, thank you for taking the time to respond. We are happy to see our answers were helpful and made you feel more positive about our work. Thanks a lot for your efforts!

---

### Official Review · Reviewer_Eg5F · 2024-07-12

**Soundness:** 2
**Presentation:** 2
**Contribution:** 2
**Rating:** 3
**Confidence:** 3

**Summary:**

This paper considers the setting of linear causal representation learning (CRL) with possibly multi-node interventions. Instead of focusing on the task of identifying the causal structure, which is recently shown to be impossible, the authors instead focus on the task of identifying mechanism shifts i.e. which nodes are intervened in each environment. The authors show that this identification task is actually possible, and design an identification algorithm to achieve it. Lastly, the authors empirically demonstrate the effectiveness of their approach.

**Strengths:**

1. The paper is well-written and easy to follow. Most mathematical definitions and statements are supported with sufficient explanations.

2. The task that the paper focuses on i.e. identifying mechanism shift is quite interesting, and can possibly be considered in other cases where full identification is hard or even impossible.

**Weaknesses:**

1. It seems that step 3 in Sec. 4.2 is stated without any proof of why it works. Is it just a heuristic or that it can provably lead to identification?

2. It seems to me that the main results of this paper is very closely related to [1], but the authors do not discuss in detail about this issue. (Please see the "Question part" for more details)

[1] Jin, Jikai, and Vasilis Syrgkanis. "Learning causal representations from general environments: Identifiability and intrinsic ambiguity." arXiv preprint arXiv:2311.12267 (2023).

**Questions:**

In the paper [1], the authors consider linear CRL (the same setting as this paper) and design an identification algorithm that works for general environments (i.e. multi-node interventions). Their algorithm can fully recover the causal graph, as well as recover the mixing matrix up to a surrounding node ambiguity (SNA). They show that SNA is am intrinsic barrier in this setting.

What I'm wondering is that, given their identification algorithm, if it is the case that the task of "identifying mechanism shift" can be straightforwardly resolved. Because if you can recover the mixing matrix, then you can also recover the noise-to-latent matrix (i.e. $B^{(k)}$) in the current paper). Then you can identify the mechanism shifts simply by comparing the entries of different $B^{(k)}$'s. Of course, the mixing matrix is actually recovered with some ambuguities. However, given that such ambiguities are inevitable as shown in [1], I suspect that such ambiguities do not affect the task of identifying mechanism shift.

*I am happy to raise my score if the above concern is appropriately addressed.*

---

> ### Author Rebuttal · Authors · 2024-08-06
>
> We thank you for your efforts in evaluating our paper. Below, we address your concerns.
>
> ### From weaknesses
>
> 1. In the population setting, step 3 will lead to identifiability provably, and it is proven in Theorem 3 that $L_i^{k,k'} = 0$ if and only if $Z_i$ is not a shifted node between environments $k$ and $k'$. However, in the finite sample setting, since we have no precise measure of the accuracy of the ICA estimation, the choice of $\alpha$ is heuristic.
>
> ### From questions
>
> * Yes, if we can estimate $B^{(k)}$ by the method in [1], SNA ambiguity does not affect the identifiability of our task. However, estimating $B^{(k)}$ up to SNA **requires** two assumptions: the number of environments $K$ should be at least equal to the number of latent nodes $d$, and there should be at least $\Theta(d^2)$ interventions. **Our algorithm does not rely on these two assumptions**. For example, for *any* number of latent nodes, our method can localize mechanism shifts even when given only $K=2$ environments.
>
> * Even if we were able to estimate $B^{(k)}$, if the objective is to identify shifted nodes, estimating less can be more efficient. Our paper shows that estimating $B^{(k)}$ is not necessary at all for achieving that objective.
>
> We will add these comments in the revision to emphasize the difference to [1].
>
> ---
> We hope our answers will make you feel more positive about our work. Please let us know any follow-up questions!

---

> > ### Comment · Reviewer_Eg5F · 2024-08-09
> >
> > I would like to thank the authors for the reply. If I understand it correctly, the authors' point is that the task of this paper is different (and strictly easier) than the one in [1], since the goal here is only to identify the shifted nodes rather than the full causal model. As a result, the assumptions required in this paper is also weaker than [1].
> >
> > Given the above interpretation of the main contributions of this paper, I would say that the task this paper considers may be of interest on its own and the results are novel. However, I'm still concerned that the contributions of this paper is a bit too close to [1]. Because although [1] requires $d$ environments (or equivalently, $d$ interventions for each node). Indeed, after completing the step 1 in their algorithm, one can directly compare the corresponding rows of $M_k$ to determine whether the node is shifted or not. Although this is not explicitly done in that paper (because their task is causal graph discovery), this deduction seems too simple to be the main result of a NeurIPS paper. Actually the main ideas underlying their algorithms seem to be the same i.e., the $i$-th row of $M_k$ implicitly encodes any information of node $i$ that is invariant under linear transformations.
> >
> > I'm also concerned about the real-world implications of the mechanism shift identification task. Indeed, without recovering the true latent nodes and the causal graph, it does not seem to be extremely useful to identify a shift at a certain node, because we have no other information of this node. How can we utilize the mechanism shift result to solve downstream tasks?

---

> > > ### Author Response · Authors · 2024-08-11
> > >
> > > Dear reviewer, thank you for participating during this discussion period, your questions are greatly appreciated. We next address them:
> > >
> > > 1. We would like to offer a different perspective on the contributions of our work. Our contributions are in explicitly *formulating, proving, and demonstrating through experiments* that one can effectively and efficiently solve the problem of identifying mechanism shifts for linear latent causal variables. Given the assumptions in the paper, we found that there is a simple yet elegant solution for this problem and we firmly consider this to be a strength, not a weakness. \
> > > Regarding the comparison to [1], it's correct that [1] also uses ICA as a first step; however, the same can be said about [2], which also uses ICA as a first step and then a couple of extra steps to identify the causal order in the fully observable setting. In fact, any other application of ICA ranging from predicting stock market prices to working with EEG data would all share with our algorithm the same step of applying ICA. This is to highlight the importance of the problem formulation, which Reviewers tbD8 and 33zq kindly appreciated from our paper.
> > >
> > > 2. Regarding the real-world implications. We believe our problem setting is more realistic, in the sense that it could be applied more widely, partly because the objective is less ambitious than estimating the full causal graph and we require less assumptions. Indeed, one of the key motivations for **directly** learning differences of causal graphs given in [3,4,5] is that learning full causal graphs is in general impractical (mainly due to strong assumptions and being sample inefficient) and, in many cases, scientists are simply interested in understanding changes among populations/distributions. Note that the latter, i.e. comparing distributions, is a fundamental question in statistics, and our setting is concerned with identifying the *latent* sources of distribution changes, which is closely related to root cause analysis, as stated in Lines 55-60.\
> > > To conclude, for example, in our real-world study in Section 5.2, the research question is whether there exists significant variations in personality traits (latent variables) across populations male/female and US/UK based on psychological tests (observed measurements). Here, our algorithm was able to answer consistently with existing psychological literature (Lines 265-266).
> > >
> > >
> > > [2] Shimizu, S., et al. (2006). A linear non-Gaussian acyclic model for causal discovery. JMLR.
> > >
> > > [3] Wang, Y., et al. (2018). Direct estimation of differences in causal graphs. NeurIPS.
> > >
> > > [4] Chen, T., et al. (2024). iSCAN: identifying causal mechanism shifts among nonlinear additive noise models. NeurIPS.
> > >
> > > [5] Malik, V., et al. (2024). "Identifying Causal Changes Between Linear Structural Equation Models." UAI.
> > >
> > >
> > > We hope these comments are helpful, we appreciate your participation during this discussion period. Please let us know any follow-up questions or concerns.

---

### Official Review · Reviewer_tbD8 · 2024-07-16

**Soundness:** 3
**Presentation:** 4
**Contribution:** 3
**Rating:** 7
**Confidence:** 4

**Summary:**

This work studies the problem of detecting the mechanism shifts in a novel way by considering the latent nodes. The authors prove identifiability results based on assumptions softer from prior identifiability results for causal representation learning. Their method is based on ICA and is evaluated empirically on synthetic data and a real-world psychometric dataset.

**Strengths:**

The paper is very well written and the scope and contribution are clearly stated and illustrated with examples.

**Related work** The related work part is detailed and correctly positions the paper at the midpoint between causal representation learning and causal mechanism shift detection.

**Novelty** The paper is novel as it proposes a methodology for a known problem (detection of causal mechanism shifts), in a new setting that considers latent variables, which is the case in the causal representation learning field.

**Theory** The authors propose and theoretically prove a softer identifiability result that allows for fewer than $d$ and unrestricted interventions (soft/hard and possibly applied to multiple nodes).

**Experiments** The method is evaluated on synthetic experiments and an interesting real-world experiment on a psychometric dataset, which provides evidence for its applicability in practice.

**Weaknesses:**

This work has some possible weaknesses

**Significance of contribution** The proposed work solves a simpler problem than causal representation learning, which doesn't require learning the whole mixing matrix $B$, but rather only the distribution shifts. I am concerned that this simplification of the problem might make it easier solvable and I wonder whether other methods of CRL could be transformed easily so they could perform well on this task.

**Experiments** Following my previous concern, I wonder why you did not compare against prior CRL methods or methods for causal mechanism shifts. Such a comparison would enhance the experimental results. Can you adapt the CRL methods or the causal mechanism shift techniques for observable variables to your setting?

**Sample complexity** From Figure 2 it seems that indeed for a large (towards infinity) number of samples, your method can accurately detect the mechanism shifts. I am concerned, however, that the sample complexity is high which makes nodes with over hundreds of nodes out of reach. This can also be seen from Table 1 where your performance drops quite early, even from graphs with 60 nodes. From a theoretical perspective, it would be interesting to compute how many samples are required regarding the number of nodes (sample complexity result - either theoretical or experimental study). Such a theorem would be appreciated by the community.

**Real experiment: size of changes** The results are very interesting and agree with psychological findings. However, the number of nodes and changes are very small. A larger number of nodes (up to 100) would be more sufficient to show that your algorithm is valid in practice.

**Limitations** You should explain either why the application to a large number of nodes is not needed in practice or add it to the limitations.

**Questions:**

Line 56: Do you think that the problem you solve is relevant to identifying the locations of the root causes of a linear SEM (as in [1])?
Figure 1 caption, 7th line: Typo, should be $Z_4\to Z_1$
Line 143: What are the implications of this assumption? Is it also present in prior identifiability results?
Line 231: What is the effect of the observed space dimension $p$ being larger or smaller with respect to the problem you solve (does it become easier or harder)?
Figure 3: the font in the legend and xticks must be larger.
Line 293-294: Here I got a bit confused. This is different from what you show in the example of Fig. 1, right? Because in Fig. 1 you show interventions (shifts) across different countries.
Line 297: Can you briefly explain how your methodology can be generalized to a nonlinear data-generating process?
Line 303: What would distribution shift imply for image data?

[1] Misiakos, P., Wendler, C., & Püschel, M. (2024). Learning DAGs from data with few root causes. Advances in Neural Information Processing Systems, 36.

**Limitations:**

The authors have discussed some limitations in the appendix. However, a significant limitation of not being applicable to a large number of nodes is not included.

---

> ### Author Rebuttal · Authors · 2024-08-06
>
> Thanks for the reviewer's recognition of our paper's novelty and contribution. We now address the reviewer's concerns.
>
> ### From Weaknesses
>
> > Significance of contributions...
>
> Please refer to the global response for this concern.
>
> > Experiments...
>
> The most recent CRL method with official code released is [2]. In their simulation, their official code is based on all Gaussian noise and samples data precision matrices from the Wishart distribution. However, in our case, our noise setting is non-Gaussian, requiring the calculation of the data precision matrix from samples. Since the estimation of the precision matrix involves the inverse of a not full-rank matrix, their code becomes numerically unstable. Thus, it is hard to accommodate their code in our setting.
>
> Instead, as you suggested, it is a good idea to compare the existing methods of directly finding causal mechanism shifts in a fully observable setting. We compared with methods DCI [3], as they are the most recent methods with official codes publicly available. The results for the simulation are shown in the table of pdf. Our method achieves higher performance in most settings.
>
> [2] Squires, C., et al. Linear causal disentanglement via interventions.
>
> [3] Wang, Y., et al. Direct estimation of differences in causal graphs
>
>
> > Sample complexity....
>
> Please refer to our gloabl response.
>
> > Real world experiment...
>
> 1. We believe that our experimental setting is relatively high-dimensional and involves larger graph sizes compared with existing CRL methods. For example, [2] considers settings where $d = 5$ latent nodes, [3] considers settings with $d = 5, 10$, and [4] considers settings where $d = 5, 8, 10$. In contrast, our experiments consider $d$ ranging from $5$ to $40$, which represents a relatively high-dimensional graph compared to existing baseline methods.
>
>     [4] Jin, J., et al. "Learning causal representations from general environments: Identifiability and intrinsic ambiguity."
>
> 2. The main contribution of our paper is to provide an identifiability result to detect the shifted nodes under fewer assumptions. Thus, we believe that the experimental part is sufficient to validate our method and proof.
>
> 3. It is challenging to find CRL datasets, under which the detected shifted nodes can also be validated by scientific papers.
>
> > Limitations
>
> Our method can be applied to any number of latent nodes, and our identifiability results holds. As we explained in previous responses, datasets with large number of latent nodes are hard to obtain, and we have already conducted synthetic experiments on relatively large latent graph sizes compared with other CRL methods. We will add more discussion about this in the limitations section of the revision.
>
> ### From Questions
>
> > Line 56...
>
> Thank you for your suggestion,we will add more discussion about it in our revision. We note that there are differences between our problem setting and the one in the mentioned paper, even in the direct observation setting. Using notations in [1], their goal is to find the SEM $A$ from the equation $X = C(I + \bar{A})$, where $(I - A)(I - \bar{A})=I$. They assume that $C$, the root cause, is sparse.
>
> In contrast, our objective is to identify the difference in $A$ across different environments, where the difference in $A$ can be sparse. The sparsity in the difference of $A$ between different environments does not necessarily imply a sparse root cause $C$. To elaborate further, suppose we have datasets from two environments, $X^1 = A^1 X^1 + N$ and $X^2 = A^2 X^2 + N$. When we take the difference, we can consider $X^1 - X^2$ as arising from the process $X = (A^1 - A^2)X + N'$. Even though $B = (A^1 - A^2)$ may be sparse, $(I + \bar{B})^{-1}$ is not necessarily sparse.
>
> Therefore, while the question addressed in that paper is relevant, it is not exactly the same as ours. We will add the citation and include a  brief discussion about the differences between our work and that paper in the revision.
>
>
> > Fig1 caption...
>
> Thanks for point this out. We will correct them in the revision.
>
> > L143...
>
> This assumption is introduced to ensure a consistent order of latent noise components after applying ICA. When ICA is applied to each individual environment, the order of noise components may differ. Therefore, this assumption helps us eliminate permutation ambiguity. However, this assumption is not necessary. We discuss how to relax this assumption in Appendix C.
>
> > L231...
>
> For any given $d$, a higher $p$ can be regarded as more data being provided since it offers additional auxiliary information. On the other hand, higher dimensional data makes optimization in ICA more challenging. Lower $p$ values provide less auxiliary information, but they are easier to optimize. Generally, we believe that higher $p$ will offer more benefits than disadvantages. For instance, in the third plot of Figure 2, we observe that higher $p$ values consistently yield higher F1 scores.
>
> > L293..
>
> Yes. Sorry for the confusion. Figure 1 and Figure 3 are two different example, and line 293-294 is based on the example of Figure 3. We will make it more clear in the revision.
>
> > L297..
>
> Please refer to the global response.
>
> > Line 303..
>
> A good example of image distribution shifts is the WILDS dataset [1]. For instance, it provides a dataset where tissue slide images  shift across patients' health conditions and hospitals. Identifying the mechanism shifts in this context involves using the tissue image dataset to determine, for example, whether some particular coloring changes across healthy and diseased patients. Since there is no evidence that the (mixing) mapping from latent to observation is linear, we did not include such an image example in our paper.
>
> [1] Koh, P., et al. "Wilds: A benchmark of in-the-wild distribution shifts." ICML 2021.
>
> ---
> We hope our answers will make you feel more positive about our work. Please let us know any follow-up questions!

---

> > ### Comment · Reviewer_tbD8 · 2024-08-07
> > **Thank you.**
> >
> > After your rebuttal, I am very positive that this is great work and I am accordingly increasing my score.
> > Particularly I appreciate the novelty of your work, by combining two distinct problems (causal representation learning and causal mechanism shifts).

---

> > > ### Author Response · Authors · 2024-08-11
> > >
> > > Dear reviewer, thank you for participating during this discussion period. We are happy to see our answers addressed your concerns and we are grateful for your support to our work. Once again, thank you.

---

### Author Rebuttal · Authors · 2024-08-06

We appreciate the time and efforts all reviewers have invested in evaluating our manuscript. We are grateful for the constructive feedback and insightful comments. It has come to our attention that there are some common questions regarding our paper, particularly concerning:

1. Can existing CRL methods be applied to our question? Is it a more difficult or simplified question than CRL?

2. Linear mixing and causal structure assumption. Can the method be generalized to nonlinear mixing functions?

3. What is the sample complexity of our method?

We next address these concerns and hope our answers will make the reviewers feel more positive about our work.

### Question 1

In some aspects, our setting does indeed simplify the CRL problem as we do not aim to identify the entire mixing matrix and latent causal structure. However, in other aspects, our setting faces other challenging scenarios such as having access to fewer than $d$ environments (even $d=2$), and considering more general interventions, such as edge weight changes, edge additions/removals, and edge reversals.

Most existing CRL methods [1,2,3] rely on each latent node having a perfect intervention *in at least* one environment. Recent developments such as [4] are capable of recovering $B^{(k)}$ under more general interventions (**none to our knowledge allow edge reversals**) but they still require *at least* $d$ environments and $\Theta(d)$ interventions. To our knowledge, no existing CRL method can address the problem of identifying latent shifted nodes in scenarios with fewer environments, fewer interventions, and under general types of interventions.

[1] Seigal, A., Squires, C. and Uhler, C. [2022], ‘Linear causal disentanglement via interventions’.

[2] Buchholz, S., Rajendran, G., Rosenfeld, E., Aragam, B., Schölkopf, B., \& Ravikumar, P. (2023). Learning linear causal representations from interventions under general nonlinear mixing. Advances in Neural Information Processing Systems, 36.

[3] Ahuja, K., Mahajan, D., Wang, Y., \& Bengio, Y. (2023). Interventional causal representation learning. In International conference on machine learning.

[4] Jin, J., \& Syrgkanis, V. (2023). Learning causal representations from general environments: Identifiability and intrinsic ambiguity. arXiv preprint arXiv:2311.12267.


### Question 2

We believe linear models are good starting points for underexplored settings such as ours, moreover, sometimes linear methods can also serve as rough approximations for nonlinear models. Currently, our method cannot be directly generalized to nonlinear mixing functions. To speculate, one way to address this could be using nonlinear ICA methods; however, a formal proof requires further research and this presents an exciting future direction.

### Question 3

The sample complexity of our algorithm is tied to the sample complexity of ICA since our algorithm uses ICA. Since there are different algorithms for solving ICA, we will assume that the estimated ICA unmixing function has the following statistical accuracy:

If $n \ge g(d,\delta)$, then with probability at least $1 - h(n,d,\delta,\epsilon)$ we have:
$$
l(\hat{M}_i - M_i) \le C \cdot p(d,n)f(\delta),
$$
where $\hat{M}_i$ is the $i$-th row of the estimated unmixing matrix $\hat{M}$. Here, $C$ is a constant, and $p$, $f$, $g$, and $h$ are known functions. For instance, in [5], $p(d,n) = \sqrt{\frac{d}{n}}$ and $f(\delta) = \sqrt{\log(1/\delta)}$. The loss function $l$ can be chosen to be the $L_2$ norm. Therefore, for two environments $k$ and $k'$, if node $i$ is not shifted:

$$
||\hat{M}_i^k - \hat{M}_i^{k'}||_2 \le ||\hat{M}_i^k - M_i||_2 + ||\hat{M}_i^{k'} - M_i||_2 \le 2 \cdot C \cdot p(d,n)f(\delta)
$$

with at least probability $1 - 2h(n,d,\delta,\epsilon)$. Thus, if we set the threshold $\alpha$ in our algorithm to $2 \cdot C \cdot p(d,n)f(\delta)$, we can control the false discovery rate to be at most $2h(n,d,\delta,\epsilon)$. A similar sample complexity theorem can be extended to more than $2$ environments with a similar technique as long as we know the sample complexity of an ICA algorithm. We will add a discussion on this in the revision.

[5] Auddy, A., \& Yuan, M. (2023). Large dimensional independent component analysis: Statistical optimality and computational tractability. arXiv preprint arXiv:2303.18156.

---

### Decision · Program_Chairs · 2024-09-25

**Decision:**

Accept (poster)

**Comment:**

This work considers linear causal representation learning and causal mechanism shifts. Reviewers praised the novelty of the work, and generally viewed it as a significant and innovative contribution to the field. I therefore recommend acceptance.